# GRU: Mitigating the Trade-off between Unlearning and Retention for LLMs

**Yue Wang** [1]  **Qizhou Wang** [1]  **Feng Liu** [2]  **Wei Huang** [3]  **Yali Du** [4]  **Xiaojiang Du** [5]  **Bo Han** [1]

## Abstract

Large language model (LLM) unlearning has demonstrated its essential role in removing privacy and copyright-related responses, crucial for their legal and safe applications. However, the pursuit of complete unlearning often comes with substantial costs due to its compromises in their general functionality, leading to a notorious trade-off between unlearning and retention. It motivates this paper to explore enhanced unlearning schemes that can mitigate this trade-off. Specifically, we propose Gradient Rectified Unlearning (GRU), an improved framework that regulates the directions of gradient updates during the unlearning procedure such that their side impacts on other, unrelated responses can be minimized. GRU is easy and general to implement, demonstrating practical effectiveness across a variety of well-established unlearning benchmarks. Our code is available at `https://github.com/tmlr-group/GRU`.

## 1. Introduction

Large language models (LLMs) (Touvron et al., 2023a; Bai et al., 2023; Liu et al., 2024a; Han et al., 2025) have revolutionized the learning paradigms towards general-purpose language generation and understanding. These models employ architectures based on multi-head attention decoders with billions of learnable parameters and are trained autoregressively on web-derived datasets containing trillions of tokens (Brown et al., 2020; Radford et al., 2021; Achiam et al., 2023). Such substantial scaling equips LLMs to tackle a wide array of complex linguistic tasks, showing remarkable capabilities across a diverse range of language tasks (Azer-

bayev et al., 2023; Roziere et al., 2023; Wu et al., 2023; Thirunavukarasu et al., 2023).

While scaling offers remarkable benefits, it also introduces substantial drawbacks. A primary concern is the propensity of LLMs to memorize data (Petroni et al., 2019; Belrose et al., 2023), potentially reproducing sensitive messages encountered during its pre-training. It encompasses copyright and privacy-related issues (Yao et al., 2023a; Liu et al., 2024b), highlighting concerns about the potential misuse of LLMs for illicit activities as well as challenges in safeguarding individual rights (Zhang et al., 2023). To remove these undesirable behaviors, it is essential to conduct regular audits to identify sensitive content and subsequently adjust the embedded knowledge within LLMs by removing them. This process is crucial for ensuring that the usage of LLMs complies with ethical and legal standards.

As the key technique to achieve this goal, LLM unlearning (Yao et al., 2023b; Liu et al., 2024b; Wang et al., 2024b) explores strategies to directly remove parameterized knowledge targeted to be unlearned. One of the foundational methods is gradient ascent (GA) (Yao et al., 2023b), which directly minimizes the log-likelihood for targeted data, thereby reducing their probabilities of being generated to nearly zero. However, GA has notably negative impacts on model responses for other, non-targeted data, spurring subsequent works that regularize unlearning procedures to retain overall model behaviors (Maini et al., 2024; Zhang et al., 2024; Wang et al., 2024a). Nevertheless, there remains an inherent trade-off between unlearning and retention, in which preserving the common performance comes at the cost of reducing the effectiveness of unlearning (Zhang et al., 2024; Liu et al., 2024b; Wuerkaixi et al., 2025; Yang et al., 2025). It motivates us to raise a pivotal research question:

> *How can we mitigate the trade-off between the process of unlearning and the goal of retaining overall performance?*

We first conduct observational experiments to better understand the model update dynamics during the unlearning process. Specifically, we delve into the fundamental component—model gradients. To do so, we separately compute the gradients of the current model on retain (non-targeted) and unlearning (targeted) data, and measure their directional

[1]TMLR Group, Department of Computer Science, Hong Kong Baptist University [2]The University of Melbourne [3]RIKEN Center for Advanced Intelligence Project [4]King's College London [5]Department of Electrical and Computer Engineering, Stevens Institute of Technology. Correspondence to: Bo Han <bhanml@comp.hkbu.edu.hk>.

*Proceedings of the 42nd International Conference on Machine Learning*, Vancouver, Canada. PMLR 267, 2025. Copyright 2025 by the author(s).

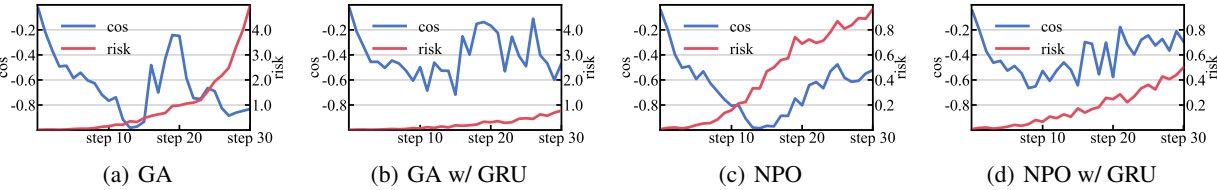

(a) GA      (b) GA w/ GRU      (c) NPO      (d) NPO w/ GRU

*Figure 1.* **Illustration of gradient dynamics during unlearning.** We visualize the cosine similarity (cos) between gradients computed on retain and unlearning data, together with the corresponding retention loss (risk), on the TOFU 5% setup. Panels (a) and (c) show GA and NPO without gradient rectification, where cosine similarity drops sharply and retention risk rises. Panels (b) and (d) show the same methods with GRU, where both curves remain stable, indicating mitigated conflict and better retention.

alignment using cosine similarity[1]. Additionally, we track the corresponding retention performance to clearly illustrate how gradient alignment affects the model's behavior throughout unlearning. In Figure 1, we present two representative pairs of visualizations, illustrating these gradient dynamics and retention performance for the representative unlearning methods GA and Negative Preference Optimization (NPO) (Zhang et al., 2024). This empirical observation motivated the design of our framework. In the following sections, we further substantiate this motivation through a formal and theoretical analysis.

To this end, we introduce the Gradient Rectified Unlearning (GRU), a general framework to mitigate the trade-off between unlearning and retention with both optimisation and geometry implications. The key insight of GRU lies in the gradient rectification during model updates: The gradients for unlearning are re-projected onto the orthogonal directions with respect to those that are detrimental to retention, thereby ensuring the overall intact performance under a first-order assumption (cf., Section 3.1). Accordingly, examples illustrating the altered behavior of gradient dynamics are shown in Figure 1(b) and (d). The directions that potentially harm retention can be estimated by the gradients from a set of data non-targeted for unlearning, which are readily accessible for many well-established benchmarks (Maini et al., 2024) or can be directly extracted from pre-trained models (Carlini et al., 2021). Please refer to Figure 2 for a conceptual illustration of our framework.

We further provide a detailed analysis to comprehend the mechanisms behind GRU. For the goal of retention, we demonstrate that GRU offers enhanced reliability over previous unlearning methods. Therein, an accurate estimation of the retention direction is crucial for its success. For the goal of unlearning, those original methods that possess gradient directions that are closer (i.e., smaller cosine similarity) to that for retention lead to better effectiveness, thereby allowing the rectified unlearning gradients to maintain a

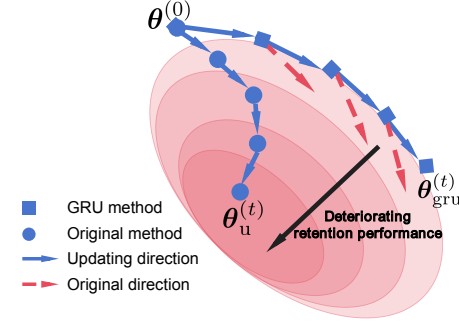

*Figure 2.* **Illustration of our unlearning method.** Conventional unlearning methods, such as GA, often suffer from declining retention performance, leading to diminished model utility. GRU mitigates this issue by rectifying the original gradients at each step, ensuring reliable unlearning without compromising retention.

substantial magnitude after adjustment. Hence, a proper choice for the basic unlearning methods is also important.

We conduct comprehensive experiments across a variety of well-established unlearning benchmarks, including TOFU (Maini et al., 2024), WMDP (Li et al., 2024), and MUSE (Shi et al., 2024). The integration of our GRU with established baselines demonstrates its effectiveness, achieving powerful unlearning capabilities alongside enhanced retention reliability. These results underscore the generality and significant potential of our approach in effectively mitigating the trade-off between unlearning and retention.

## 2. Preliminaries

We consider a pre-trained LLM that models an autoregressive distribution over sequences of tokens. Specifically, for an input sequence $\boldsymbol{s} = [s_1, s_2, \ldots, s_{|\boldsymbol{s}|}]$, the probability of the sequence is modeled as the product of conditional probabilities of each token given all preceding tokens:

$$p(\boldsymbol{s}; \boldsymbol{\theta}) = \prod_{i=1}^{|\boldsymbol{s}|} p(s_i \mid \boldsymbol{s}_{1:i-1}; \boldsymbol{\theta}),$$

---

[1]We refer to data that are not targeted for unlearning as "retain data" and data targeted to be unlearned as "unlearning data," aligning with existing literature (Maini et al., 2024).

where $\boldsymbol{\theta}$ denotes model parameters, and $\boldsymbol{s}_{1:i-1}$ represents the subsequence consisting of tokens $s_1$ through $s_{i-1}$. $\boldsymbol{\theta}$ is typically learned by minimizing the negative log-likelihood (NLL) loss over a large corpus of web-sourced data $\mathcal{D}_{\mathrm{t}} = \{\boldsymbol{s}^1, \boldsymbol{s}^2, \ldots, \boldsymbol{s}^m\}$ of size $m$, which is given by $-1/m \sum_{\boldsymbol{s} \in \mathcal{D}_{\mathrm{t}}} \log p(\boldsymbol{s}; \boldsymbol{\theta})$. Pre-trained LLMs have shown their remarkable capabilities (Zhao et al., 2023). However, these models also face safety concerns due to their reliance on web-sourced data, potentially leading to privacy breaches (Das et al., 2024), copyright infringement (Eldan & Russinovich, 2023), and potential misuse (Yao et al., 2024).

## 2.1. LLM Unlearning

These concerns motivate the emerging studies of LLM unlearning recently, which aims to effectively remove undesirable data points or entire hazardous domains from the original models. Formally speaking, let $\mathcal{D}_{\mathrm{u}} = \{\boldsymbol{s}_{\mathrm{u}}^1, \boldsymbol{s}_{\mathrm{u}}^2, \ldots, \boldsymbol{s}_{\mathrm{u}}^n\}$ represents the unlearning dataset, typically a subset of the training data $\mathcal{D}_{\mathrm{t}}$ where $n \ll m$. The primary objectives of LLM unlearning are twofold (Liu et al., 2024b):

a) **Removal**: The unlearned model, characterized by parameters $\boldsymbol{\theta}_{\mathrm{u}}$, should eliminate the knowledge associated with $\mathcal{D}_{\mathrm{u}}$, thereby reducing its capacity to recall or reproduce any information targeted to be forgotten.

b) **Retention**: The model should also retain its performance on the remaining data $\mathcal{D}_{\mathrm{t}} \setminus \mathcal{D}_{\mathrm{u}}$, ensuring that the capabilities on tasks and data unrelated to the unlearning dataset can be preserved in reliable manner.

The objectives of removal and retention are both essential for LLM unlearning, which can be interpreted as a bi-objective learning problem (Liu et al., 2024b; Wang et al., 2024b).

## 2.2. Unlearning Methods

In the following, we present several representative methods for unlearning, each addressing ways to remove or preserve retention performance, while striving to mitigate the trade-off between the two goals. We further discuss gradient projection, a foundational strategy in machine learning, and its recent advances in addressing competing objectives.

**Gradient ascent (GA)** is one of the most fundamental unlearning methods, which minimizes the log-likelihood for targeted data. The unlearning objective of GA is

$$\min_{\boldsymbol{\theta}} \mathcal{L}_{\mathrm{GA}}(\mathcal{D}_{\mathrm{u}}; \boldsymbol{\theta}) \coloneqq \frac{1}{n} \sum_{\boldsymbol{s} \in \mathcal{D}_{\mathrm{u}}} \log p(\boldsymbol{s}; \boldsymbol{\theta}), \qquad (1)$$

which directly reduces the probabilities of generating contents resembling $\mathcal{D}_{\mathrm{u}}$ to approach zero, thereby leading to effective knowledge removal. However, due to its extremely large strengths of gradient updates, the resulting

GA-unlearned models will suffer from excessive unlearning (Liu et al., 2024b), where the model responses for non-targeted data will also be damaged, i.e., GA is not good at retention. It motivates a series of subsequent works to improve the retention performance for the resulting models.

**Gradient Difference (GD)** regularizes GA with a retain dataset $\mathcal{D}_{\mathrm{r}}$ of size $m'$, typically sampled from $\mathcal{D}_{\mathrm{t}} \setminus \mathcal{D}_{\mathrm{u}}$ and $m' \ll m$. These data represent the knowledge that should be preserved. The associated retain loss, which is given by

$$\mathcal{R}(\mathcal{D}_{\mathrm{r}}; \boldsymbol{\theta}) = -\frac{1}{m'} \sum_{\boldsymbol{s} \in \mathcal{D}_{\mathrm{r}}} \log p(\boldsymbol{s}; \boldsymbol{\theta}), \qquad (2)$$

serves as regularization in conjunction with GA, namely,

$$\begin{aligned} \min_{\boldsymbol{\theta}} &\ \mathcal{L}_{\mathrm{GD}}(\mathcal{D}_{\mathrm{u}}, \mathcal{D}_{\mathrm{r}}; \boldsymbol{\theta}) \\ &\coloneqq \mathcal{L}_{\mathrm{GA}}(\mathcal{D}_{\mathrm{u}}; \boldsymbol{\theta}) + \lambda \mathcal{R}(\mathcal{D}_{\mathrm{r}}; \boldsymbol{\theta}), \end{aligned} \qquad (3)$$

where $\lambda$ is a trade-off hyper-parameter, typically set to 1. However, many previous works (Maini et al., 2024) reveal that the unlearning term, i.e., $\mathcal{L}_{\mathrm{GA}}(\mathcal{D}_{\mathrm{u}}; \boldsymbol{\theta})$, tends to dominate the dynamics of gradient updates. Therefore, GD may still strongly impact retention performance negatively.

**Negative Preference Optimization (NPO)** (Zhang et al., 2024) directly refines the objective of GA to mitigate excessive unlearning, of which the formulation is motivated by direct preference optimization, a well-known preference alignment method (Rafailov et al., 2024). NPO segregates the dis-preferred part from DPO, heuristically employing it as the unlearning objective, following the formulation of

$$\begin{aligned} \min_{\boldsymbol{\theta}} &\ \mathcal{L}_{\mathrm{NPO}}(\mathcal{D}_{\mathrm{u}}; \boldsymbol{\theta}) \\ &\coloneqq \frac{1}{n} \sum_{\boldsymbol{s} \in \mathcal{D}_{\mathrm{u}}} \frac{2}{\beta} \log \left[ 1 + \left( \frac{p(\boldsymbol{s}; \boldsymbol{\theta})}{p(\boldsymbol{s}; \boldsymbol{\theta}_{\mathrm{org}})} \right)^{\beta} \right], \end{aligned} \qquad (4)$$

where $\beta$ is the inverse temperature and $\boldsymbol{\theta}_{\mathrm{org}}$ denotes model parameters before unlearning. The effects of NPO in mitigating excessive unlearning can be understood through its gradients, which are equivalent to GA with extra reweighting (Zhang et al., 2024). This weighting mechanism pays more attention to data that have small impacts on retention. However, the strength of unlearning for NPO is weaker than that for GA, which could lead to inadequate unlearning.

**Unlearning with Control (UWC)** (Wang et al., 2024a) suggests a post-unlearning calibration framework. UWC blends model parameters from before and after unlearning to restore retention performance. With a meticulous-searched controlling parameter $\alpha$, we have the calibrated model of

$$\alpha \boldsymbol{\theta}_{\mathrm{u}} + (1 - \alpha) \boldsymbol{\theta}_{\mathrm{org}}, \qquad (5)$$

whose performance on $\mathcal{D}_{\mathrm{t}} \setminus \mathcal{D}_{\mathrm{f}}$ can approach that of $\boldsymbol{\theta}_{\mathrm{org}}$. UWC is flexible in integration with various unlearning methods, while its ability to address excessive unlearning still comes at the cost of compromising the effects of unlearning.

**Gradient Rectification for Conflicting Goals.** The idea of modifying gradient directions to resolve conflicts between competing objectives has been explored in various domains, including continual learning and multi-task learning (Lopez-Paz & Ranzato, 2017; Yu et al., 2020). This idea was formalized in continual learning by Gradient Episodic Memory (GEM) (Lopez-Paz & Ranzato, 2017), which constrains the current task's update so that it does not increase the loss on past tasks, using a quadratic program to project the update direction into the feasible region defined by gradients of previous tasks. Subsequently, similar geometric principles were adopted in multi-task learning, Gradient Surgery for Multi-Task Learning (PCGrad) (Yu et al., 2020) detects gradient conflicts between tasks and projects each task's gradient to reduce destructive interference. Our work extends the reach of gradient projection methods, providing a theoretical and practical framework tailored to the specific challenge, i.e., the trade-off between unlearning and retention in LLMs.

# 3. Gradient Rectified Unlearning

As discussed above, many methods have been developed to mitigate excessive unlearning. However, these achievements often result in an inevitable trade-off between removal and retention—improvements in maintaining the overall performance typically occur at the expense of weakened strength of unlearning. This trade-off is detrimental to practical LLM unlearning, since both the goals of removal and reliable retention are essential: Compromising on removal risks privacy breaches and harmful behaviors; compromising on retention can adversely affect the overall utility of the model, negatively affecting its commercial value.

In this paper, rather than developing new methods that can better balance the trade-off between removal and retention, we turn our focus toward directly breaking this dichotomy. In other words, we aim to explore frameworks in which improved unlearning does not compromise the overall utility.

## 3.1. Motivation and The Proposed Framework

In this section, we formalize our goal towards avoiding trade-offs by studying a constrained gradient updating rule.

To begin with, considering any unlearning objective $\mathcal{R}_{\mathrm{u}}$ mentioned in Section 2, we recall the conventional stochastic updating rule at the $t$-th step in the following:

$$\boldsymbol{\theta}^{(t+1)} \leftarrow \boldsymbol{\theta}^{(t)} - \mathtt{lr} \cdot \boldsymbol{g}_{\mathrm{u}}^{(t)}. \tag{6}$$

Therein, $\mathtt{lr}$ denotes the (un) learning rate and $\boldsymbol{g}_{\mathrm{u}}^{(t)} = \nabla_{\boldsymbol{\theta}} \mathcal{L}(\tilde{\mathcal{D}}_{\mathrm{u}}^{(t)}; \boldsymbol{\theta}^{(t)})$ with $\tilde{\mathcal{D}}_{\mathrm{u}}^{(t)}$ the mini-batch of size $b$ sampled from $\mathcal{D}_{\mathrm{u}}$ and $\mathcal{L}$ being any unlearning loss mentioned in Section 2, e.g., GA, GD, or NPO. This direct updating rule has proven to be unreliable in terms of retention, leading

to the undesirable trade-off between retention and removal, which is widely mentioned in many previous works (Wang et al., 2024a; Liu et al., 2024b; Maini et al., 2024).

This notable drawback motivates us to replace $\boldsymbol{g}_{\mathrm{u}}^{(t)}$ in Eq. (6) with its constrained version $\tilde{\boldsymbol{g}}_{\mathrm{u}}^{(t)}$: We incorporate the retain loss $\mathcal{R}$ as in Eq. (2), along with the corresponding gradients $\boldsymbol{g}_{\mathrm{r}}^{(t)} = \nabla_{\boldsymbol{\theta}} \mathcal{R}(\mathcal{D}_{\mathrm{r}}; \boldsymbol{\theta}^{(t)})$, typically estimated by random mini-batch drawn from $\mathcal{D}_{\mathrm{r}}$. Then, we assert that the adjusted gradients $\tilde{\boldsymbol{g}}_{\mathrm{u}}^{(t)}$ should meet the condition as

$$\begin{aligned} \underset{\tilde{\boldsymbol{g}}_{\mathrm{u}}^{(t)}}{\arg\min} \quad & \|\tilde{\boldsymbol{g}}_{\mathrm{u}}^{(t)} - \boldsymbol{g}_{\mathrm{u}}^{(t)}\|^2 \\ \text{s.t.} \quad & \langle \tilde{\boldsymbol{g}}_{\mathrm{u}}^{(t)}, \boldsymbol{g}_{\mathrm{r}}^{(t)} \rangle \geq 0. \end{aligned} \tag{7}$$

The objective $\min \|\tilde{\boldsymbol{g}}_{\mathrm{u}}^{(t)} - \boldsymbol{g}_{\mathrm{u}}^{(t)}\|^2$ ensures that the constrained gradients remain close to their original values. Meanwhile, the constraint $\langle \tilde{\boldsymbol{g}}_{\mathrm{u}}^{(t)}, \boldsymbol{g}_{\mathrm{r}}^{(t)} \rangle \geq 0$ guarantees that the updates will not impair the model performance on retain data. Overall, Eq. (7) encapsulates our principle that the removal of targeted knowledge should occur under strict conditions that ensure the retention of performance on non-targeted data, thereby mitigating the inherent trade-off. As we mitigate the trade-off by adjusting the gradient direction, we name the corresponding unlearning framework as **gradient rectified unlearning (GRU)**.

The rationale behind $\langle \tilde{\boldsymbol{g}}_{\mathrm{u}}^{(t)}, \boldsymbol{g}_{\mathrm{r}}^{(t)} \rangle \geq 0$ for retention is simple: Assuming the model is locally linear (Wortsman et al., 2022), we can approximate the expected loss change for the retain data as $\mathcal{R}(\boldsymbol{\theta} + \mathtt{lr} \cdot \tilde{\boldsymbol{g}}_{\mathrm{u}}) - \mathcal{R}(\boldsymbol{\theta}) \approx -\mathtt{lr} \langle \tilde{\boldsymbol{g}}_{\mathrm{u}}^{(t)}, \boldsymbol{g}_{\mathrm{r}}^{(t)} \rangle$. As observed, a positive $\langle \tilde{\boldsymbol{g}}_{\mathrm{u}}^{(t)}, \boldsymbol{g}_{\mathrm{r}}^{(t)} \rangle$ implies that the loss $\mathcal{R}$ does not deteriorate following gradient updates, thereby ensuring the goal of retention. Later, we will show that the condition expressed in Eq. (7) remains valid under some less stringent assumptions, further highlighting its practical applicability.

## 3.2. Realizations

This section explores details to implement GRU, focusing on its closed-form solution as well as additional strategies to enhance its reliability in practice.

**Closed-form solution**. Eq. (7) is a constrained optimization problem that is not easy to be implemented. However, it constitutes a quadratic programming problem with a linear constraint, allowing us to derive its closed-form solution. Specifically, the adjustment gradients can be written as:

$$\begin{aligned} \tilde{\boldsymbol{g}}_{\mathrm{u}}^{(t)} = & \boldsymbol{g}_{\mathrm{u}}^{(t)} + \frac{\max(-\langle \boldsymbol{g}_{\mathrm{u}}^{(t)}, \boldsymbol{g}_{\mathrm{r}}^{(t)} \rangle, 0)}{\|\boldsymbol{g}_{\mathrm{r}}^{(t)}\|^2} \boldsymbol{g}_{\mathrm{r}}^{(t)} \\ = & \boldsymbol{g}_{\mathrm{u}}^{(t)} + \frac{\|\boldsymbol{g}_{\mathrm{u}}^{(t)}\| \max(-\cos(\boldsymbol{g}_{\mathrm{u}}^{(t)}, \boldsymbol{g}_{\mathrm{r}}^{(t)}), 0)}{\|\boldsymbol{g}_{\mathrm{r}}^{(t)}\|} \boldsymbol{g}_{\mathrm{r}}^{(t)}, \end{aligned} \tag{8}$$

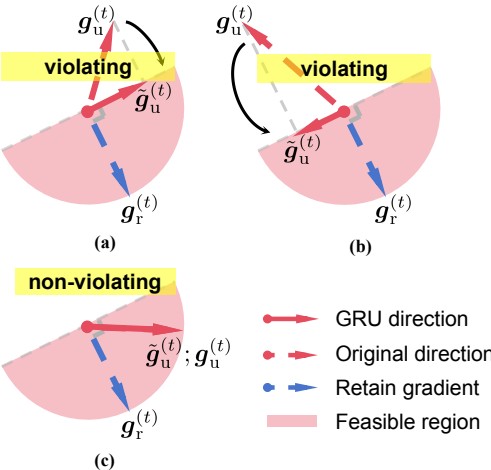

*Figure 3.* **Illustration of GRU Updating Rule.** Panels (a)-(b) display situations where the angles between the gradient vectors (red dashed and blue dashed arrows) are obtuse, violating the constraint in Eq. (7). In these cases, the gradients should be adjusted orthogonally. Panel (c) illustrates a scenario with an acute angle between the gradient vectors, adhering to the constraint in Eq. (7) and falling within the retention-safe feasible region (red half-circle), thus requiring no further adjustment.

where $\cos(\boldsymbol{g}_{\mathrm{u}}^{(t)}, \boldsymbol{g}_{\mathrm{r}}^{(t)}) = \langle \boldsymbol{g}_{\mathrm{u}}^{(t)}, \boldsymbol{g}_{\mathrm{r}}^{(t)} \rangle / (\|\boldsymbol{g}_{\mathrm{u}}^{(t)}\| \cdot \|\boldsymbol{g}_{\mathrm{r}}^{(t)}\|)$. For more detailed derivations, please refer to Appendix A.1.

Eq. (8) conveys a clear geometric interpretation, adjusting the original gradients $\boldsymbol{g}_{\mathrm{u}}^{(t)}$ onto the half-space defined by the constraint $\langle \boldsymbol{g}_{\mathrm{u}}^{(t)}, \boldsymbol{g}_{\mathrm{r}}^{(t)} \rangle \geq 0$. If this constraint is already satisfied, then $\boldsymbol{g}_{\mathrm{u}}^{(t)}$ remains unchanged. Otherwise, $\boldsymbol{g}_{\mathrm{u}}^{(t)}$ will be projected in the direction that is orthogonal to $\boldsymbol{g}_{\mathrm{r}}^{(t)}$. Please refer to Figure 3 for some visual illustrations. Moreover, we present the implementation of our GRU in Algorithm 1, further elaborating on several key details as follows.

**Stable Estimation**. In practice, we typically utilize stochastic mini-batches of data to estimate the exact values of $\boldsymbol{g}_{\mathrm{r}}^{(t)}$ outlined in Eq. (7). Specifically, as shown in Algorithm 1, the mini-batches $\mathcal{B}_{\mathrm{u}}^{(t)}$ and $\mathcal{B}_{\mathrm{r}}^{(t)}$ serve as substitutes for the complete datasets $\mathcal{D}_{\mathrm{u}}$ and $\mathcal{D}_{\mathrm{r}}$. However, this may introduce stochastic errors, particularly when the batch size is small, which is commonly the case in LLM unlearning. Therefore, we employ the exponential moving average (EMA) to mitigate the additional computation costs associated with increasing batch sizes, namely,

$$\bar{\boldsymbol{g}}_{\mathrm{r}}^{(t)} = (1 - \gamma)\bar{\boldsymbol{g}}_{\mathrm{r}}^{(t-1)} + \gamma \boldsymbol{g}_{\mathrm{r}}^{(t)}, \qquad (9)$$

where $\gamma \in [0, 1)$ is the smoothing parameter, with smaller values suggesting that a broader range of recent batches is covered, indicating a large batch size implicitly. It is worth noting that EMA is an approximation of using large batch sizes, given that $\boldsymbol{\theta}$ itself is also updated throughout the steps

---

**Algorithm 1** GRU Framework

1: **Input:** Initial parameters $\boldsymbol{\theta}_{\mathrm{org}}$, learning rate `lr`, number of iterations $T$, and hyperparameters $\gamma, \tau$.
2: Initialize $\bar{\boldsymbol{g}}_{\mathrm{r}}^{(0)} = \boldsymbol{0}$;
3: **for** $t = 0, 1, \ldots, T - 1$ **do**
4:     sample the mini-batches of $\mathcal{B}_{\mathrm{u}}^{(t)}$ and $\mathcal{B}_{\mathrm{r}}^{(t)}$ from $\mathcal{D}_{\mathrm{u}}$ and $\mathcal{D}_{\mathrm{r}}$, respectively;
5:     $\boldsymbol{g}_{\mathrm{u}}^{(t)} \leftarrow \nabla_{\boldsymbol{\theta}} \mathcal{L}(\mathcal{B}_{\mathrm{u}}^{(t)}; \boldsymbol{\theta}^{(t)})$;
6:     $\boldsymbol{g}_{\mathrm{r}}^{(t)} \leftarrow \nabla_{\boldsymbol{\theta}} \mathcal{R}_{\mathrm{r}}(\mathcal{B}_{\mathrm{r}}^{(t)}; \boldsymbol{\theta}^{(t)})$;
7:     $\bar{\boldsymbol{g}}_{\mathrm{r}}^{(t)} \leftarrow (1 - \gamma)\bar{\boldsymbol{g}}_{\mathrm{r}}^{(t-1)} + \gamma \boldsymbol{g}_{\mathrm{r}}^{(t)}$;
8:     **if** $\langle \boldsymbol{g}_{\mathrm{u}}^{(t)}, \bar{\boldsymbol{g}}_{\mathrm{r}}^{(t)} \rangle < 0$ **then**
9:         $\tilde{\boldsymbol{g}}_{\mathrm{u}}^{(t)} \leftarrow \boldsymbol{g}_{\mathrm{u}}^{(t)} - \frac{\langle \boldsymbol{g}_{\mathrm{u}}^{(t)}, \bar{\boldsymbol{g}}_{\mathrm{r}}^{(t)} \rangle}{\|\bar{\boldsymbol{g}}_{\mathrm{r}}^{(t)})\|^2} \bar{\boldsymbol{g}}_{\mathrm{r}}^{(t)}$;
10:    **else**
11:        $\tilde{\boldsymbol{g}}_{\mathrm{u}}^{(t)} \leftarrow \boldsymbol{g}_{\mathrm{u}}^{(t)}$;
12:    **end if**
13:    **if** $\|\tilde{\boldsymbol{g}}_{\mathrm{u}}^{(t)}\| > \tau$ **then**
14:        $\tilde{\boldsymbol{g}}_{\mathrm{u}}^{(t)} \leftarrow \tau \tilde{\boldsymbol{g}}_{\mathrm{u}}^{(t)} / \|\tilde{\boldsymbol{g}}_{\mathrm{u}}^{(t)}\|$;
15:    **end if**
16:    $\boldsymbol{\theta}^{(t+1)} \leftarrow \boldsymbol{\theta}^{(t)} - \mathrm{lr}^{(t)} \tilde{\boldsymbol{g}}_{\mathrm{u}}^{(t)}$;
17: **end for**
18: Return $\boldsymbol{\theta}^{(T)}$.

---

$t$. Therefore, selecting the appropriate value for $\gamma$ is crucial, as it involves balancing the representation of a larger batch size against minimizing the induced errors.

**Gradient Clipping**. Due to stochastic variations and low-order approximations, the rectified gradients may inadvertently encroach upon regions that may decrease retention. To further enhance the practical reliability of our GRU, we further constrain the gradient norm via gradient clipping, following many previous works such as (Wortsman et al., 2022; Wang et al., 2024a). Specifically, the gradients are scaled down to ensure it stays within a bounded range, i.e.,

$$\tilde{\boldsymbol{g}}_{\mathrm{u}}^{(t)} \leftarrow \begin{cases} \tilde{\boldsymbol{g}}_{\mathrm{u}}^{(t)}, & \text{if } \|\tilde{\boldsymbol{g}}_{\mathrm{u}}^{(t)}\| \leq \tau \\ \tau \tilde{\boldsymbol{g}}_{\mathrm{u}}^{(t)} / \|\tilde{\boldsymbol{g}}_{\mathrm{u}}^{(t)}\|, & \text{if } \|\tilde{\boldsymbol{g}}_{\mathrm{u}}^{(t)}\| > \tau \end{cases}, \qquad (10)$$

where $\tau$ is the predefined threshold for the maximal-allowed value for the norm of the rectified gradients.

### 3.3. Theoretical Analysis

In this section, we present formal analyses to further substantiate the efficacy of our GRU, which focuses on two main aspects: **a) Efficacy in Removal**: In Theorem 3.1, we demonstrate the convergence of the GRU updating dynamics for unlearning. **b) Reliability in Retention**: In Theorem 3.2, we illustrate that our GRU is capable to preserve overall model performance, surpassing the cases without GRU. Overall, we formally verify that our GRU can mitigate the notorious trade-off between removal and retention, thus ensuring overall superior unlearning efficacy.

We begin by showing that unlearning with GRU ensures convergence in lie with the original objective of unlearning.

**Theorem 3.1.** *Assume the unlearning objective $\mathcal{L}$ is differentiable, $L$-smooth, and lower bounded. Then, the GRU update rule with the learning rate $\text{lr} < 2/L$ will converge to either a) a degenerate configuration where $\cos\big(\boldsymbol{g}_{\text{u}}^{(t)}, \boldsymbol{g}_{\text{r}}^{(t)}\big) = -1$ at a specific step $t$, or b) the locally optimal solution $\boldsymbol{\theta}^*$ that minimizes $\mathcal{L}(\mathcal{D}_{\text{u}}; \boldsymbol{\theta})$.*

*Remark.* Overall, Theorem 3.1 demonstrates that, from a convergence perspective, the GRU does not compromise the original goal of unlearning. This is contingent upon avoiding those cases where $\cos\big(\boldsymbol{g}_{\text{u}}^{(t)}, \boldsymbol{g}_{\text{r}}^{(t)}\big) = -1$. Moreover, given that stochastic optimization is employed for LLM unlearning, we can simply overcome this issue by randomly selecting a new data batch from the unlearning dataset, thereby allowing the unlearning process to continue. Please refer to Appendix A.2 for the detailed proof.

Moreover, central to our motivation, we justify that our GRU can better maintain model performance on non-targeted data compared to original unlearning rules without GRU.

**Theorem 3.2.** *Assume that the retain loss $\mathcal{R}$ is differentiable and $L$-smooth, and the $\text{lr}$-curvature $\mathfrak{H}_{\text{lr}}(\mathcal{R}; \boldsymbol{g})$ for $\mathcal{R}$ (cf., Definition A.1) satisfies $\mathfrak{H}_{\text{lr}}(\mathcal{R}; \boldsymbol{g}) \geq \ell \|\boldsymbol{g}\|^2$ for any gradients $\boldsymbol{g}$ and some constant $\ell \leq L$. Let $\boldsymbol{\theta}_{\text{gru}}^{(t+1)}$ and $\boldsymbol{\theta}_{\text{u}}^{(t+1)}$ be the parameters after applying one step of gradient updates for the original $\boldsymbol{\theta}^{(t)}$ with and without GRU, respectively. Then, we can ensure $\mathcal{R}(\mathcal{D}_{\text{r}}; \boldsymbol{\theta}_{\text{gru}}^{(t+1)}) \leq \mathcal{R}(\mathcal{D}_{\text{r}}; \boldsymbol{\theta}_{\text{u}}^{(t+1)})$ if a) $\ell \geq L\Big(1 - \langle \boldsymbol{g}_{\text{u}}^{(t)}, \boldsymbol{g}_{\text{r}}^{(t)} \rangle^2 / (\|\boldsymbol{g}_{\text{u}}^{(t)}\|^2 \|\boldsymbol{g}_{\text{r}}^{(t)}\|^2)\Big)$ and b) $0 < \text{lr} \leq \frac{2}{L}$.*

*Remark.* In heuristics, $1 - \langle \boldsymbol{g}_{\text{u}}^{(t)}, \boldsymbol{g}_{\text{r}}^{(t)} \rangle^2 / (\|\boldsymbol{g}_{\text{u}}^{(t)}\|^2 \|\boldsymbol{g}_{\text{r}}^{(t)}\|^2 = \sin^2 \phi$ quantifies the degree of conflict between $\boldsymbol{g}_{\text{u}}^{(t)}$ and $\boldsymbol{g}_{\text{r}}^{(t)}$; larger values (i.e., gradients closer to orthogonal) generally indicate a greater potential to harm the retain performance. Hence, condition a) implies that, when the conflict is more severe, our requirement on the curvature ratio $\ell/L$ must be correspondingly stronger. Condition b) is the classical stability constraint $0 < \text{lr} \leq 2/L$ for gradient descent on an $L$-smooth function, ensuring the validity of the quadratic bound adopted in GRU. Please refer to Appendix A.3 for the detailed proof.

Overall, Theorem 3.1 ensures that GRU will not compromise convergence for the original unlearning objective, and Theorem 3.2 further characterizes its behaviors in preserving the overall model performance. Taken together, we certify the efficacy of our GRU in mitigating the notorious trade-off between removal and retention.

## 4. Go Beyond GRU

Most unlearning methods, including our GRU, rely on retain data to preserve the overall performance. However, the retain data adopted in current benchmarks can often exhibit distributional bias. For example, in the TOFU setup, specific author profiles are selectively unlearned while the remaining profiles are retained. Yet, the broader objective of retention is to preserve model capacity across a diverse range of domains, such as the humanities, sciences, and general knowledge. As a result, the current retain data may not be fully representative, with bias arising from the distributional shift between the adopted retain set and the broader expected data distribution encountered in real-world applications (Huang et al., 2023). It motivates us to investigate a challenging scenario where we need rely exclusively on the unlearn data $\mathcal{D}_{\text{u}}$, without further access to the retain data $\mathcal{D}_{\text{r}}$. To adapt for this setup, we make several adjustments for GRU and propose **task vector rectified unlearning (TRU)**.

The key insight behind TRU is that unlearning typically involves a series of data points rather than a single instance. Thus, for each individual data point $\boldsymbol{s}_{\text{u}} \in \mathcal{D}_{\text{u}}$ targeted for unlearning, the remaining data points within $\mathcal{D}_{\text{u}}$, i.e., $\mathcal{D}_{\text{u}} \setminus \{\boldsymbol{s}_{\text{u}}\}$, can offer information for retention if used properly. Here, we incorporate the so-called task vectors (Ilharco et al., 2022), which is critical in our algorithmic design.

**Task Vector.** A task vector typically represents the necessary adjustments for model parameters to incorporate new knowledge. For example, when we want the model to learn from a specific data point $\boldsymbol{s}$, we initiate by fine-tuning the current model parameterized, denoted by $\boldsymbol{\theta}_{\text{org}}$. It can be achieved through $T$ iterations of gradient updates, following $\boldsymbol{\theta}^{(t+1)} = \boldsymbol{\theta}^{(t)} + \text{lr} \cdot \nabla_{\boldsymbol{\theta}} \log p(\boldsymbol{s}; \boldsymbol{\theta}^{(t)})$ with $\boldsymbol{\theta}^{(0)} = \boldsymbol{\theta}_{\text{org}}$ and $\boldsymbol{\theta}_{\boldsymbol{s}} = \boldsymbol{\theta}^{(T)}$. Obviously, $\boldsymbol{T}_{\boldsymbol{s}}$ allows for the augmentation of the original model with the knowledge acquired from $\boldsymbol{s}$ by applying $\boldsymbol{\theta}_{\text{org}} + \boldsymbol{T}_{\boldsymbol{s}}$. Conversely, to unlearn a data point $\boldsymbol{s}_{\text{u}}$, we can reverse this process by subtracting the task vector via $\boldsymbol{\theta}_{\text{org}} - \boldsymbol{T}_{\boldsymbol{s}_{\text{u}}}$ (Barbulescu & Triantafillou, 2024).

**Rectified Task Vector.** However, the task vector $\boldsymbol{T}_{\boldsymbol{s}_{\text{u}}}$ still faces the trade-off between unlearning and retention. To address this, we begin by considering a simple scenario to remove a single data point $\boldsymbol{s}_{\text{u}}$. Hence, a similar constrained updating rule, as outlined in Eq. (7), can be adopted and further adjusted as:

$$\min_{\tilde{\boldsymbol{T}}_{\boldsymbol{s}_{\text{u}}}} \|\tilde{\boldsymbol{T}}_{\boldsymbol{s}_{\text{u}}} - \boldsymbol{T}_{\boldsymbol{s}_{\text{u}}}\|^2$$
$$\text{s.t. } \langle -\tilde{\boldsymbol{T}}_{\boldsymbol{s}_{\text{u}}}, \nabla_{\boldsymbol{\theta}} \mathcal{R}(\mathcal{D}_{\text{u}} \setminus \{\boldsymbol{s}_{\text{u}}\}; \boldsymbol{\theta}_{\text{org}}) \rangle \geq 0. \tag{11}$$

It mandates that the task vector be rectified to have no negative impact on other data points. For this purpose, we utilize the internal reference set for retention, $\mathcal{D}_{\text{u}} \setminus \{\boldsymbol{s}_{\text{u}}\}$, to construct a rectified task vector for $\boldsymbol{s}_{\text{u}}$. Similar to Eq. (8), we

have its closed-form solution as

$$\tilde{\boldsymbol{T}}_{\boldsymbol{s}_{\mathrm{u}}} = \boldsymbol{T}_{\boldsymbol{s}_{\mathrm{u}}}$$
$$+ \frac{[\langle \boldsymbol{T}_{\boldsymbol{s}_{\mathrm{u}}}, \nabla_{\boldsymbol{\theta}} \mathcal{R}(\mathcal{D}_{\mathrm{u}} \setminus \{\boldsymbol{s}_{\mathrm{u}}\}; \boldsymbol{\theta}_{\mathrm{org}}) \rangle]_-}{\|\nabla_{\boldsymbol{\theta}} \mathcal{R}(\mathcal{D}_{\mathrm{u}} \setminus \{\boldsymbol{s}_{\mathrm{u}}\}; \boldsymbol{\theta}_{\mathrm{org}})\|^2} \boldsymbol{T}_{\boldsymbol{s}_{\mathrm{u}}}. \quad (12)$$

This mechanism naturally extends to multiple data points, where we compute a rectified task vector $\tilde{\boldsymbol{T}}s\mathrm{u}$ for each $\boldsymbol{s}\mathrm{u} \in \mathcal{D}\mathrm{u}$. Moreover, to ensure standardised influence across data points, these task vectors are further normalized so that their magnitudes are equal to 1. The resulting normalized vectors are denoted as $\bar{\boldsymbol{T}}_{\boldsymbol{s}_{\mathrm{u}}}$, facilitating equitable integration across data points. The final unlearning update is then formed by aggregating (e.g., averaging) these rectified vectors across all elements in $\mathcal{D}_{\mathrm{u}}$:

$$\boldsymbol{\theta}_{\mathrm{org}} - \frac{\mathtt{stg}}{n} \sum_{\boldsymbol{s}_{\mathrm{u}} \in \mathcal{D}_{\mathrm{u}}} \bar{\boldsymbol{T}}_{\boldsymbol{s}_{\mathrm{u}}}, \quad (13)$$

where we subtract the average of all normalized task vectors from the original model and $n$ represents the number of data points within $\mathcal{D}_{\mathrm{u}}$ and $\mathtt{stg}$ indicates the strength of task vector-based unlearning. It ensures a reliable removal of targeted data while mitigating the compromise to the overall performance. Notably, because each rectified task vector is constructed with respect to its own reference subset, they remain mutually compatible, and their aggregation yields a stable and robust overall unlearning direction. Moreover, when the number of data points $n$ is substantial, an effective strategy involves randomly dividing the entire unlearning set into several smaller batches. Each batch then serves as a substitute for $\boldsymbol{s}_{\mathrm{u}}$ in Eq. (11), reducing the demands associated with calculating the task vectors.

## 5. Experiments

In this section, we conduct extensive experiments to verify the effectiveness of our GRU in mitigating the trade-off involved in LLM unlearning. To begin with, we first offer a brief description of our experimental setups.

**Benchmarks.** Our evaluations adopt three representative benchmarks: TOFU (Maini et al., 2024), WMDP (Li et al., 2024), and MUSE (Shi et al., 2024). TOFU comprises 200 synthetic author profiles, totally 4,000 question-answer pairs. It covers different unlearning setups with varying proportions of data targeted to be unlearned, including 1%, 5%, or 10% of the profiles as unlearning sets. WMDP collects a set of sensitive knowledge encountered in practice, further categorized into three areas as biosecurity, cybersecurity, and chemical security. MUSE constructs their unlearning sets using news articles and books, primarily focusing on addressing copyright issues within existing LLMs.

**Baselines and Backbones.** For the baseline methods, we focus on a set of objective-based approaches, including GA,

GD, NPO, weighted gradient ascent (WGA) (Wang et al., 2024b). All of these methods have demonstrated their practical significance and are thus adopted in our experiments. Moreover, for the backbone models, we adhere to the default suggestions for each benchmarks. We use further fine-tuned LLaMA2-7B-chat (Touvron et al., 2023b) and Phi-1.5 (Abdin et al., 2024) for TOFU; Zephyr-7B-beta (Tunstall et al., 2023) for WMDP; ICLM-7B (Shi et al., 2023) for MUSE.

**Hyper-parameters Configurations.** In our experiments, We employ the AdamW optimizer (Loshchilov & Hutter, 2017) with the batch size of 32 and the learning rates $2 \times 10^{-5}$ for Phi-1.5 and $1 \times 10^{-5}$ for LLaMA2-7B-chat in TOFU; $1 \times 10^{-5}$ in MUSE; $4 \times 10^{-6}$ in WMDP. Furthermore, the training steps are set to 5 epochs for TOFU, 1 epoch for MUSE, and 20 steps for WMDP. For the hyperparameters within GRU, we employ grid search on validation data to identify their optimal values. The candidate values for $\gamma$ include $\{0.01, 0.05, 0.1, 0.2, 0.3, 0.4, 0.5, 0.6, 0.7, 0.8, 0.9, 0.99\}$, and that for $\tau$ are $\{0.001, 0.005, 0.01, 0.1, 1.0, 10, 100\}$. Their specific choices and their impacts across different baseline methods are detailed in Appendix D.

**Metrics.** We adhere to the suggested evaluation metrics for each benchmark. TOFU adopts two metrics: FQ and MU. FQ measures the extent of data removal by the statistical difference in responses between unlearned models and ground-standard models, which are trained without targeted data. Higher values of FQ are preferred, and we report the logarithm of the original FQ values for enhanced readability. MU assesses the overall performance of retention, which is a combination of several foundational metrics. It can be computed on the retain sets, real authors, and world facts, where higher values indicate better retention.

WMDP performs QA evaluations on WMDP-Bio and WMDP-Cyber to assess the efficacy of removal, where the prompts are standardized following (Gao et al., 2024). For retention, WMDP also utilizes QA evaluations, but conducting on the MMLU benchmark. Therein, smaller values of QA evaluations are preferred for WMDP-Bio and WMDP-Cyber, while larger values are desired for MMLU. Moreover, MUSE proposes two metrics to assess the removal efficacy, i.e., VerbMem and KnowMem, quantifying various aspects of memorization and membership inference. MUSE also uses KnowMem for assessing performance retention, where larger values are preferred. To ease analysis, we use the symbols ↑ and ↓ next to metric names to indicate that their larger/smaller values are preferred.

**Hardware Configurations.** All our experiments are conducted with a series of computation nodes powered by NVIDIA-A100-80GB GPUs and Intel(R) Xeon(R) Gold 6248R CPUs. All our codes are implemented on Transformers version 4.42.4 and CUDA version 12.1.

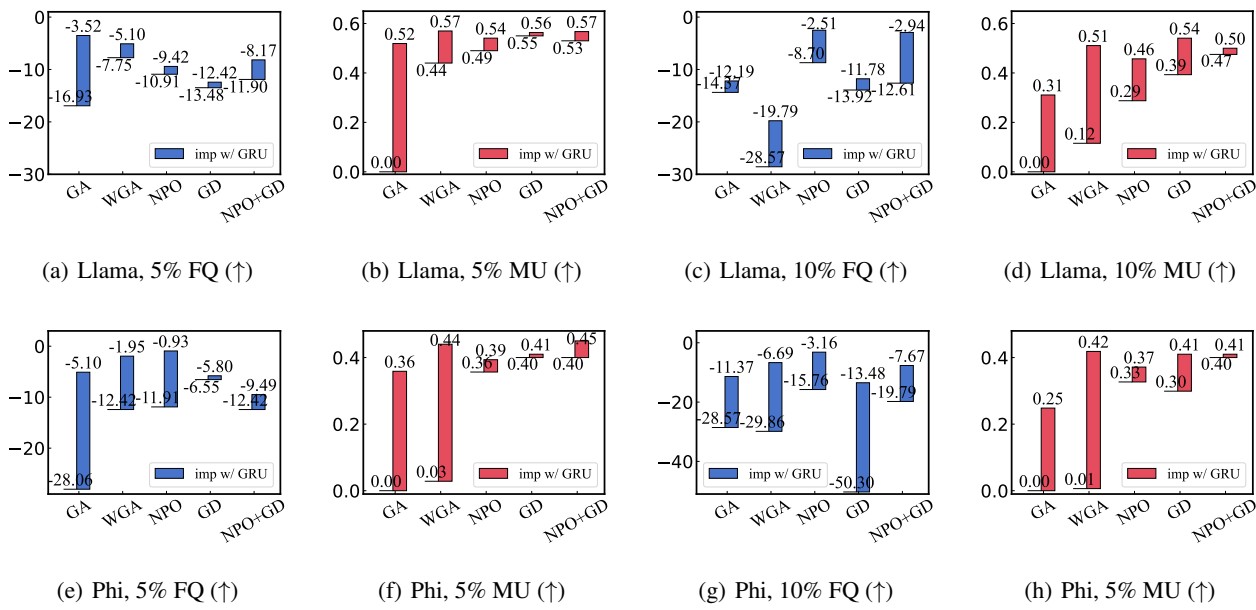

(a) Llama, 5% FQ (↑)   (b) Llama, 5% MU (↑)   (c) Llama, 10% FQ (↑)   (d) Llama, 10% MU (↑)

(e) Phi, 5% FQ (↑)   (f) Phi, 5% MU (↑)   (g) Phi, 10% FQ (↑)   (h) Phi, 5% MU (↑)

*Figure 4.* Experimental results on the TOFU benchmarks: Evaluating 5% and 10% unlearning setups using Llama-2-7B (Llama) and Phi-1.5 (Phi) backbones. We present metric scores—either FQ or MU— both without and with GRU, displayed in pairs and highlighted the corresponding improvements after GRU (imp w/ GRU) with colored bars. For example, in (a), the FQ values of -16.93 (w/o GRU) and -3.52 (w/ GRU) for GA are connected by a blue-colored bar, signifying the improvements attributed to GRU.

## 5.1. Main Results

GRU is a general framework compatible with a wide range of objective-based unlearning methods. In this section, we demonstrate its reliability by integrating it with various unlearning approaches. Our goal is to show the universal improvements achieved with GRU across different methods in both removal and retention, thereby justifying the overall efficacy of our GRU in mitigating their trade-off.

**TOFU Benchmark.** We consider five representative baseline methods—GA, WGA, NPO, GD, and NPO+GD—to validate their performance improvements after implementing GRU in terms of both removal (FQ) and retention (MU) metrics. We summarize our experimental results in Figure 4, focusing on the challenging setups of 5% and 10% unlearning. Additional experimental setups and baseline methods are detailed in Appendix B. We observe uniform improvements in both FQ and MU metrics after applying GRU, across various methods, unlearning setups, and backbone models. Surprisingly, even for methods typically viewed as less promising, such as GA, we observe significant enhancements in both removal and retention after incorporating GRU. The improvements observed in other methods, such as WGA and GD, are also very impressive.

On the other side, with the integration of GRU, it remains difficult to identify a single baseline method that always outperforms others across different unlearning scenarios

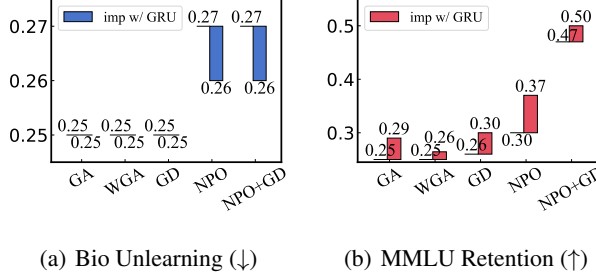

(a) Bio Unlearning (↓)   (b) MMLU Retention (↑)

*Figure 5.* Experimental results on the WMDP benchmarks with QA accuracies evaluated on Bio unlearning and MMLU, quantifying the efficacy of removal and retention, respectively.

and setups. For example, with Phi-1.5, WGA and NPO are more effective than others. When coming to Llama-2-7B, GA and WGA tend to be more suitable under the 5% unlearning setup, whereas NPO and NPO+GD show greater efficacy under a 10% unlearning setup. Thus, while GRU uniformly enhances the overall efficacy of unlearning, the selection of baseline methods remains a task-dependent consideration that requires careful selection.

**WMDP and MUSE Benchmarks.** To further substantiate the general efficacy and reliability of our GRU, we conduct additional experiments using the WMDP and MUSE benchmarks, of which the results are detailed in Figure 5

*Table 1.* Experimental results on the TOFU benchmarks within the retain-data-free settings are presented. We compare our TRU with representative baseline models, across different unlearning setups and backbone architectures. The top-performing results in each column are highlighted in bold to ease reference.

| Method | Phi-1.5 | | | | Llama2-7B | | | |
|---|---|---|---|---|---|---|---|---|
| | 5% | | 10% | | 5% | | 10% | |
| | FQ↑ | MU↑ | FQ↑ | MU↑ | FQ↑ | MU↑ | FQ↑ | MU↑ |
| original | -28.84 | 0.52 | -40.52 | 0.52 | -32.13 | 0.63 | -48.59 | 0.63 |
| retrain | 0.00 | 0.52 | 0.00 | 0.53 | 0.00 | 0.60 | 0.00 | 0.61 |
| TV | -46.18 | 0.00 | -36.06 | 0.00 | -22.13 | 0.00 | -9.06 | 0.00 |
| GA | -28.06 | 0.00 | -28.57 | 0.00 | -16.93 | 0.00 | -14.37 | 0.00 |
| WGA | -12.42 | 0.03 | -29.86 | 0.01 | -7.75 | 0.44 | -28.57 | 0.12 |
| NPO | -11.91 | 0.36 | -15.76 | 0.33 | -10.91 | 0.49 | -8.70 | 0.29 |
| TRU | **-9.04** | **0.40** | **-13.04** | **0.36** | **-7.34** | **0.53** | **-4.92** | **0.47** |

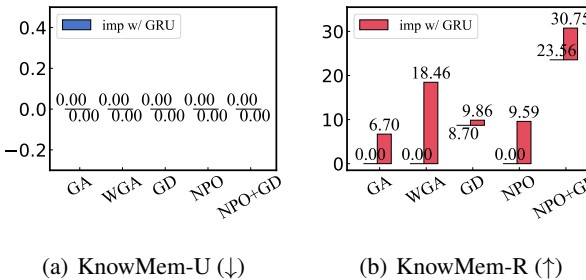

(a) KnowMem-U (↓)    (b) KnowMem-R (↑)

*Figure 6.* Experimental results on the MUSE benchmarks with KnowMem, assessing the efficacy of removal and retention on targeted and non-targeted data, respectively.

and Figure 6, respectively. Note that the minimum values for QA accuracy and KnowMem are 0.25 and 0, and thus the results shown for GA and GD in Figure 5(a) and for all methods in Figure 6 cannot decrease further.

Overall, our results demonstrate that GRU remains reliable across various baseline methods and unlearning setups, enhancing the overall efficacy of unlearning with notable improvements or maintenance in both the goals of data removal and retention. Additionally, it is evident that the NPO-based methods generally deliver superior performance. Given these observations, which can be recommended as our default choices for effective unlearning.

## 5.2. Retain Data Free

We further consider the retain-data-free settings, where we have no retain data at hand as mentioned in Section 4. As a case study, we test the efficacy of various methods that do not rely on retain data and our TRU. We further include the baseline of task vector (TV) (Ilharco et al., 2022) for fair comparison, which is also the key technique that is adopted in our TRU. The experimental results are summarized in Table 1, where we also report metric scores for the model before unlearning (original) as well as for the gold standard model (retrain), which is fine-tuned from scratch without the targeted data. Across baseline methods, it can be observed that the retain-data-free settings is challenging. Only WGA and NPO can demonstrate some ability of reliable unlearning. In contrast, other methods, such as TV and GA, can render the unlearned models completely useless. Furthermore, our TRU exhibits notable improvements over these baselines in both removal and retention, showcasing the broad applications of our unlearning schemes suggested in Eq. (7) even in some more restricted unlearning setups.

## 5.3. More Results in Appendices

Due to space limit, we leave more detailed experimental results and additional analyses to the appendices. For convenience, this section provides a brief overview of these contents: In Appendix B, we offer more comprehensive results for our main experiments on varying benchmarks and metrics, further covering other baselines such as SimNPO (Fan et al., 2024) and NPO+KL. Additionally, we include a comparison with RMU, the method proposed alongside WMDP, specifically evaluated on the WMDP benchmark together with its combination with GRU (see Appendix C for results and discussion). In Appendix D, we perform a hyper-parameter sensitivity analysis and outline their recommended setups. Finally, in Appendix E, we include our ablation studies and other experimental analyses. Finally, we provide a more practically Meaningful and fairer comparative analysis in Section F.

## 6. Conclusion

This paper introduces GRU, a novel and general framework designed to mitigate the inherent trade-off between data removal and retention for LLM unlearning, a critical challenge in this field. Our key insight involves regulating the gradients used for unlearning by projecting them onto the orthogonal complement of directions that negatively affect retention. Thereby, GRU ensures that the unlearning updates minimize their adverse impact on the overall model performance. We offer both theoretical analyses and empirical evidence to demonstrate the effectiveness of our method in mitigating the trade-off between removal and retention, resulting in overall efficacy of unlearning. However, our method critically relies on the quality of retain data. While TRU can mitigate this issue to some extent, potential biases and distribution shifts therein may still be detrimental. In the future, we will explore ways to pursue reliable LLM unlearning without relying on retain data or their surrogates.

## Impact Statement

This work has notable societal implications by spurring the development of LLMs that meet both ethical and legal standards, thus mitigating the risks of privacy breaches and the unauthorized spread of protected information. We advocate for continued research into legally sound and reliable LLMs that honor individual rights and intellectual property while maintaining their robustness across numerous applications. By paving the way for broader deployment of LLMs capable of adapting to evolving legal and ethical requirements, this line of work helps ensure that the broader adoption of LLMs that can adapt to evolving legal and ethical standards remains both trustworthy and socially beneficial.

## Acknowledgements

YW, QZW, and BH were supported by HKBU Faculty Niche Research Areas No. RC-FNRA-IG/22-23/SCI/04 and HKBU CSD Departmental Incentive Scheme. FL was supported by the Australian Research Council (ARC) with grant number DE240101089, LP240100101, DP230101540 and the NSF&CSIRO Responsible AI program with grant number 2303037. The authors also would like to express their sincere gratitude to the anonymous reviewers and the area chairs for their thorough review and constructive feedback. Their insightful comments and valuable suggestions have significantly enhanced the quality and clarity of this manuscript. We deeply appreciate their effort in helping us improve our work.

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

# A. Derivations and Proofs

To begin with, we present our derivations regarding the closed-form solutions for Eq. (7) as well as detailed proofs for theoretical analyses in Section 3.3.

## A.1. The Closed-form Solution of Eq. (7)

Recalling the original optimization problem of

$$\underset{\tilde{g}_{\mathrm{u}}^{(t)}}{\arg\min} \quad \|\tilde{g}_{\mathrm{u}}^{(t)} - g_{\mathrm{u}}^{(t)}\|^2$$
$$\text{s.t.} \qquad \langle \tilde{g}_{\mathrm{u}}^{(t)}, g_{\mathrm{r}}^{(t)} \rangle \geq 0$$

We construct the Lagrangian equation following

$$\frac{1}{2}\|\tilde{g}_{\mathrm{u}}^{(t)} - g_{\mathrm{u}}^{(t)}\|^2 - \kappa \langle \tilde{g}_{\mathrm{u}}^{(t)}, g_{\mathrm{r}}^{(t)} \rangle \tag{14}$$

with $\kappa \geq 0$ the Lagrange multiplier. Then, setting the gradients Eq. (14) with respect to $\tilde{g}_{\mathrm{u}}^{(t)}$ to zero, we have

$$\tilde{g}_{\mathrm{u}}^{(t)} - g_{\mathrm{u}}^{(t)} - \kappa g_{\mathrm{r}}^{(t)} = 0. \tag{15}$$

It indicates that we have the solution of

$$\tilde{g}_{\mathrm{u}}^{(t)} = g_{\mathrm{u}}^{(t)} + \kappa g_{\mathrm{r}}^{(t)}.$$

Substituting it back into the constraint:

$$\langle \tilde{g}_{\mathrm{u}}^{(t)}, g_{\mathrm{r}}^{(t)} \rangle = \langle g_{\mathrm{u}}^{(t)} + \kappa g_{\mathrm{r}}^{(t)}, g_{\mathrm{r}}^{(t)} \rangle = \langle g_{\mathrm{u}}^{(t)}, g_{\mathrm{r}}^{(t)} \rangle) + \kappa \|g_{\mathrm{r}}^{(t)}\|^2 \geq 0$$

and solving for $\kappa$, we have

$$\kappa \geq -\frac{\langle g_{\mathrm{u}}^{(t)}, g_{\mathrm{r}}^{(t)} \rangle}{\|g_{\mathrm{r}}^{(t)}\|^2}.$$

Since $\kappa \geq 0$, we can further derive

$$\kappa = \frac{[-\langle g_{\mathrm{u}}^{(t)}, g_{\mathrm{r}}^{(t)} \rangle]_+}{\|g_{\mathrm{r}}^{(t)}\|^2} = \frac{[\langle g_{\mathrm{u}}^{(t)}, g_{\mathrm{r}}^{(t)} \rangle]_-}{\|g_{\mathrm{r}}^{(t)}\|^2}.$$

Then, we can obtain the closed-form for the adjusted gradients as

$$\tilde{g}_{\mathrm{u}}^{(t)} = g_{\mathrm{u}}^{(t)} + \frac{[\langle g_{\mathrm{u}}^{(t)}, g_{\mathrm{r}}^{(t)} \rangle]_-}{\|g_{\mathrm{r}}^{(t)}\|^2} g_{\mathrm{r}}^{(t)}$$

Thus, we complete our derivation for Eq. (7).

We further demonstrate that the GRU causes the magnitudes of the rectified gradients to decrease. To this end, we first decompose the original gradients $g_{\mathrm{u}}^{(t)}$ into two orthogonal components that are parallel and perpendicular to $g_{\mathrm{r}}^{(t)}$, which are

$$g_{\mathrm{u}}^{(t)} = g_\perp + g_\parallel \text{ and } g_\perp \perp g_{\mathrm{r}}^{(t)}.$$

with $g_\parallel$ parallel to $g_{\mathrm{r}}^{(t)}$. In this decomposition, $g_\parallel$ represents the component of $g_{\mathrm{u}}^{(t)}$ that aligns with $g_{\mathrm{r}}^{(t)}$, whereas $g_\perp$ is orthogonal to $g_{\mathrm{r}}^{(t)}$. Then, if $\langle g_{\mathrm{u}}^{(t)}, g_{\mathrm{r}}^{(t)} \rangle \geq 0$, no adjustment is needed and we keep $\tilde{g}_{\mathrm{u}}^{(t)} = g_{\mathrm{u}}^{(t)}$. In this case, the norm remains the same as $\|\tilde{g}_{\mathrm{u}}^{(t)}\| = \|g_{\mathrm{u}}^{(t)}\|$. However, when $\langle g_{\mathrm{u}}^{(t)}, g_{\mathrm{r}}^{(t)} \rangle < 0$, $g_\parallel$ represents a negatively aligned component with respect to $g_{\mathrm{r}}^{(t)}$. The correction term in Eq. (8) removes this negative parallel portion, thereby setting $\tilde{g}_{\mathrm{u}}^{(t)} = g_\perp$. Since $g_{\mathrm{u}}^{(t)} = g_\perp + g_\parallel$, we have $\|g_{\mathrm{u}}^{(t)}\|^2 = \|g_\perp\|^2 + \|g_\parallel\|^2$. When the parallel component is negative relative to $g_{\mathrm{r}}^{(t)}$, its removal decreases the overall norm. Thus, it is easy to conclude that, after rectification, we have

$$\|\tilde{g}_{\mathrm{u}}^{(t)}\| = \|g_\perp\| < \|g_{\mathrm{u}}^{(t)}\|.$$

This difference in magnitudes is influenced by the angles between $g_{\mathrm{u}}^{(t)}$ and $g_{\mathrm{r}}^{(t)}$. Overall, as these angles widen beyond $90°$, the magnitudes of the negative component $g_\parallel$ increases. It indicates that a greater portion of this components should be removed to fulfill the constraint specified in Eq. (7). In an extreme case where the angles approach $180°$, nearly the entire $g_{\mathrm{u}}^{(t)}$ is inverted relative to $g_{\mathrm{r}}^{(t)}$. It implies a substantial reduction in magnitudes of $g_{\mathrm{u}}^{(t)}$ to approach 0 after adjustment.

## A.2. Proof of Theorem 3.1

*Proof.* To simply our symbology, we use $\mathcal{L}(\boldsymbol{\theta})$ and $\mathcal{R}(\boldsymbol{\theta})$ to replace $\mathcal{L}(\mathcal{D}_\mathrm{u}; \boldsymbol{\theta})$ and $\mathcal{R}(\mathcal{D}_\mathrm{r}; \boldsymbol{\theta})$ if raising no confusion.

By the $L$-smoothness of $\mathcal{L}$, we have

$$\mathcal{L}(\boldsymbol{\theta}^{(t)} - \mathrm{lr}\,\tilde{\boldsymbol{g}}_\mathrm{u}^{(t)}) \;\leq\; \mathcal{L}(\boldsymbol{\theta}^{(t)}) \;-\; \mathrm{lr}\,\boldsymbol{g}_\mathrm{u}^{(t)\top}\tilde{\boldsymbol{g}}_\mathrm{u}^{(t)} \;+\; \frac{L\,\mathrm{lr}^2}{2}\,\|\tilde{\boldsymbol{g}}_\mathrm{u}^{(t)}\|^2.$$

If we further define

$$\Delta^{(t)} \;:=\; -\,\mathrm{lr}\,\boldsymbol{g}_\mathrm{u}^{(t)\top}\tilde{\boldsymbol{g}}_\mathrm{u}^{(t)} \;+\; \frac{L\,\mathrm{lr}^2}{2}\,\|\tilde{\boldsymbol{g}}_\mathrm{u}^{(t)}\|^2.$$

Then, if $\Delta^{(t)} < 0$, we have a strict decrease, i.e., $\mathcal{L}(\boldsymbol{\theta}^{(t+1)}) < \mathcal{L}(\boldsymbol{\theta}^{(t)})$, and thus we can complete our proof. As observed, $\Delta^{(t)}$ consists of two terms:

- **Linear term:** Since $\tilde{\boldsymbol{g}}_\mathrm{u}^{(t)}$ is a projection of $\boldsymbol{g}_\mathrm{u}^{(t)}$ that does not invert the direction, we have $\boldsymbol{g}_\mathrm{u}^{(t)\top}\tilde{\boldsymbol{g}}_\mathrm{u}^{(t)} \geq 0$.

- **Quadratic term:** Since the norm of $\tilde{\boldsymbol{g}}_\mathrm{u}^{(t)}$ is bounded by $\|\boldsymbol{g}_\mathrm{u}^{(t)}\|$, we have $\frac{L\mathrm{lr}^2}{2}\|\tilde{\boldsymbol{g}}_\mathrm{u}^{(t)}\|^2 \leq \frac{L\mathrm{lr}^2}{2}\|\boldsymbol{g}_\mathrm{u}^{(t)}\|^2$.

Hence the sign of $\Delta^{(t)}$ depends on the term of $-\,\mathrm{lr}\,+\,\frac{L\mathrm{lr}^2}{2}$, where we need to ensure

$$-\,\mathrm{lr} + L\mathrm{lr}^2/2 < 0 \quad\Longrightarrow\quad \mathrm{lr} < 2/L.$$

Under the above condition, the negative linear term dominates the quadratic penalty term, so we have $\Delta^{(t)} < 0$ and

$$\mathcal{L}(\boldsymbol{\theta}^{(t+1)}) \;<\; \mathcal{L}(\boldsymbol{\theta}^{(t)}).$$

Thus, we obtain a strict descent unless we encounter a degenerate scenario. Specifically, if $\boldsymbol{g}_\mathrm{u}^{(t)}$ happens to be exactly reversed with respect to the retain gradients $\boldsymbol{g}_\mathrm{r}^{(t)}$, i.e. their angle is $180°$ and $\cos(\boldsymbol{g}_\mathrm{u}^{(t)}, \boldsymbol{g}_\mathrm{r}^{(t)}) = -1$, then, after rectification, one obtains $\tilde{\boldsymbol{g}}_\mathrm{u}^{(t)} = 0$. In this case, we have $\boldsymbol{\theta}^{(t+1)} = \boldsymbol{\theta}^{(t)}$, which makes no further decrease. Thus, we complete the proof. $\qquad\square$

## A.3. Proof of Theorem 3.2

*Proof.* Before giving the detailed proof, we first provide the following definition of the $q$-curvature.

**Definition A.1** ($q$-Curvature). For any smooth and differentiable loss $\mathcal{R}$, its $q$-Curvature with respect to some gradients $\boldsymbol{g}$ is defined as

$$\mathfrak{H}_q(\mathcal{R}; \boldsymbol{g}) \;=\; \int_0^1 (1-a)\left[\boldsymbol{g}^\top \nabla^2 \mathcal{R}\big(\boldsymbol{\theta}^{(t)} - a\,q\,\boldsymbol{g}\big)\,\boldsymbol{g}\right] da. \tag{16}$$

which quantifies the curvature of the local optimization landscape, with larger values indicating a sharper loss landscape.

Recall that at the $t$-th iteration, the original updating rule without GRU is $\boldsymbol{\theta}_\mathrm{u}^{(t+1)} = \boldsymbol{\theta}_\mathrm{u}^{(t)} - \mathrm{lr}\,\boldsymbol{g}_\mathrm{u}^{(t)}$. Additionally, according to the integral form of Taylor theorem, for any $\alpha \in [0, 1]$, we have

$$\mathcal{R}(\boldsymbol{\theta}^{(t)} - \mathrm{lr}\,\boldsymbol{g}_\mathrm{u}^{(t)}) = \mathcal{R}(\boldsymbol{\theta}^{(t)}) + \int_0^1 \nabla\mathcal{R}(\boldsymbol{\theta}^{(t)} - a\,\mathrm{lr}\,\boldsymbol{g}_\mathrm{u}^{(t)})^\top[-\mathrm{lr}\,\boldsymbol{g}_\mathrm{u}^{(t)}]\,da.$$

Separating the first-order (linear) portion and the second-order (Hessian) portion, one can write:

$$\mathcal{R}(\boldsymbol{\theta}_\mathrm{u}^{(t+1)}) = \mathcal{R}(\boldsymbol{\theta}^{(t)}) \;-\; \mathrm{lr}\,\langle\boldsymbol{g}_\mathrm{r}^{(t)}, \boldsymbol{g}_\mathrm{u}^{(t)}\rangle \;+\; \frac{1}{2}\int_0^1 [-\mathrm{lr}\,\boldsymbol{g}_\mathrm{u}^{(t)}]^\top \nabla^2 \mathcal{R}\big(\boldsymbol{\theta}^{(t)} - a\,\mathrm{lr}\,\boldsymbol{g}_\mathrm{u}^{(t)}\big)[-\mathrm{lr}\,\boldsymbol{g}_\mathrm{u}^{(t)}]\,da.$$

Since we assume $\mathfrak{H}_{\mathrm{lr}}(\mathcal{R}; \boldsymbol{g}_\mathrm{u}^{(t)}) \geq \ell\,\|\boldsymbol{g}_\mathrm{u}^{(t)}\|^2$, we have

$$\int_0^1 [-\mathrm{lr}\,\boldsymbol{g}_\mathrm{u}^{(t)}]^\top \nabla^2 \mathcal{R}(\boldsymbol{\theta}^{(t)} - a\,\mathrm{lr}\,\boldsymbol{g}_\mathrm{u}^{(t)})[-\mathrm{lr}\,\boldsymbol{g}_\mathrm{u}^{(t)}]\,da \;\geq\; \ell\,\mathrm{lr}^2\,\|\boldsymbol{g}_\mathrm{u}^{(t)}\|^2,$$

and thus

$$\mathcal{R}(\boldsymbol{\theta}^{(t)} - \mathtt{lr}\, \boldsymbol{g}_{\mathrm{u}}^{(t)}) \;\geq\; \mathcal{R}(\boldsymbol{\theta}^{(t)}) \;-\; \mathtt{lr}\, \langle \boldsymbol{g}_{\mathrm{r}}^{(t)},\, \boldsymbol{g}_{\mathrm{u}}^{(t)} \rangle \;+\; \frac{\ell\, \mathtt{lr}^2}{2}\, \|\boldsymbol{g}_{\mathrm{u}}^{(t)}\|^2, \tag{17}$$

which establishes the lower bound for $\mathcal{R}(\boldsymbol{\theta}_{\mathrm{u}}^{(t+1)}) = \mathcal{R}(\boldsymbol{\theta}^{(t)} - \mathtt{lr}\, \boldsymbol{g}_{\mathrm{u}}^{(t)})$. For the rectified updating rule with GRU, due to the $L$-smoothness, we have

$$\mathcal{R}(\boldsymbol{\theta}_{\mathrm{gru}}^{(t+1)}) \;\leq\; \mathcal{R}(\boldsymbol{\theta}^{(t)}) \;-\; \mathtt{lr}\, \langle \boldsymbol{g}_{\mathrm{r}}^{(t)},\, \tilde{\boldsymbol{g}}_{\mathrm{u}}^{(t)} \rangle \;+\; \frac{L\, \mathtt{lr}^2}{2}\, \|\tilde{\boldsymbol{g}}_{\mathrm{u}}^{(t)}\|^2. \tag{18}$$

Combining Eq. (17) and Eq. (18), we have

$$\Delta = \mathcal{R}(\boldsymbol{\theta}_{\mathrm{u}}^{(t+1)}) \;-\; \mathcal{R}(\boldsymbol{\theta}_{\mathrm{gru}}^{(t+1)})$$

$$\geq \underbrace{\left[ \mathcal{R}(\boldsymbol{\theta}^{(t)}) \;-\; \mathtt{lr}\, \langle \boldsymbol{g}_{\mathrm{r}}^{(t)},\, \boldsymbol{g}_{\mathrm{u}}^{(t)} \rangle \;+\; \frac{\ell\, \mathtt{lr}^2}{2}\, \|\boldsymbol{g}_{\mathrm{u}}^{(t)}\|^2 \right]}_{\text{Lower bound for } \mathcal{R}(\boldsymbol{\theta}_{\mathrm{u}}^{(t+1)})} \;-\; \underbrace{\left[ \mathcal{R}(\boldsymbol{\theta}^{(t)}) \;-\; \mathtt{lr}\, \langle \boldsymbol{g}_{\mathrm{r}}^{(t)},\, \tilde{\boldsymbol{g}}_{\mathrm{u}}^{(t)} \rangle \;+\; \frac{L\, \mathtt{lr}^2}{2}\, \|\tilde{\boldsymbol{g}}_{\mathrm{u}}^{(t)}\|^2 \right]}_{\text{Upper bound for } \mathcal{R}(\boldsymbol{\theta}_{\mathrm{gru}}^{(t+1)})}.$$

After organizing, we have

$$\Delta \;\geq\; \left[ -\mathtt{lr}\, \langle \boldsymbol{g}_{\mathrm{r}}^{(t)},\, \boldsymbol{g}_{\mathrm{u}}^{(t)} \rangle \;+\; \tfrac{\ell\, \mathtt{lr}^2}{2}\, \|\boldsymbol{g}_{\mathrm{u}}^{(t)}\|^2 \right] \;-\; \left[ -\mathtt{lr}\, \langle \boldsymbol{g}_{\mathrm{r}}^{(t)},\, \tilde{\boldsymbol{g}}_{\mathrm{u}}^{(t)} \rangle \;+\; \tfrac{L\, \mathtt{lr}^2}{2}\, \|\tilde{\boldsymbol{g}}_{\mathrm{u}}^{(t)}\|^2 \right]$$

$$= \underbrace{-\mathtt{lr}\, \langle \boldsymbol{g}_{\mathrm{r}}^{(t)},\, \boldsymbol{g}_{\mathrm{u}}^{(t)} \rangle \;+\; \mathtt{lr}\, \langle \boldsymbol{g}_{\mathrm{r}}^{(t)},\, \tilde{\boldsymbol{g}}_{\mathrm{u}}^{(t)} \rangle}_{\text{(linear-difference term)}} \;+\; \underbrace{\frac{\ell\, \mathtt{lr}^2}{2}\, \|\boldsymbol{g}_{\mathrm{u}}^{(t)}\|^2 \;-\; \frac{L\, \mathtt{lr}^2}{2}\, \|\tilde{\boldsymbol{g}}_{\mathrm{u}}^{(t)}\|^2}_{\text{(quadratic-difference term)}}.$$

Now, we show that the formulations inside each bracket is non-negative:

1. **Rectification Nonnegativity.** Since $\tilde{\boldsymbol{g}}_{\mathrm{u}}^{(t)}$ is formed from $\boldsymbol{g}_{\mathrm{u}}^{(t)}$ by removing negatively aligned components with respect to $\boldsymbol{g}_{\mathrm{r}}^{(t)}$, we have $\langle \boldsymbol{g}_{\mathrm{r}}^{(t)},\, \tilde{\boldsymbol{g}}_{\mathrm{u}}^{(t)} \rangle \geq \langle \boldsymbol{g}_{\mathrm{r}}^{(t)},\, \boldsymbol{g}_{\mathrm{u}}^{(t)} \rangle$, and thus $-\langle \boldsymbol{g}_{\mathrm{r}}^{(t)},\, \boldsymbol{g}_{\mathrm{u}}^{(t)} \rangle + \langle \boldsymbol{g}_{\mathrm{r}}^{(t)},\, \tilde{\boldsymbol{g}}_{\mathrm{u}}^{(t)} \rangle \geq 0$. Multiplication by $\mathtt{lr}$ preserves non-negativity, ensuring that the expression inside the first bracket remains non-negative.

2. **Curvature condition.** By construction $\|\tilde{\boldsymbol{g}}_{\mathrm{u}}^{(t)}\| = \|\boldsymbol{g}_{\mathrm{u}}^{(t)}\| \sin \phi$, where $\phi$ is the angle between $\boldsymbol{g}_{\mathrm{u}}^{(t)}$ and $\boldsymbol{g}_{\mathrm{r}}^{(t)}$. Condition **a)** of the theorem states $\ell \geq L(1 - \cos^2 \phi) = L \sin^2 \phi$. Therefore

$$\ell \|\boldsymbol{g}_{\mathrm{u}}^{(t)}\|^2 - L \|\tilde{\boldsymbol{g}}_{\mathrm{u}}^{(t)}\|^2 = \|\boldsymbol{g}_{\mathrm{u}}^{(t)}\|^2 (\ell - L \sin^2 \phi) \;\geq\; 0,$$

implying the quadratic-difference term is non–negative for every $0 < \mathtt{lr} \leq 2/L$ (condition **b)**).

Since both terms are non–negative, we have $\Delta \geq 0$, i.e., $\mathcal{R}(\boldsymbol{\theta}_{\mathrm{gru}}^{(t+1)}) \leq \mathcal{R}(\boldsymbol{\theta}_{\mathrm{u}}^{(t+1)})$. $\qquad \square$

# B. Detailed Results

This section provides comprehensive results that echo our main experiments discussed in Section 5.1. It encompasses TOFU, WMDP, and MUSE benchmarks, further incorporating additional baseline methods like SimNPO, and other metrics, such as PrivLeak for MUSE. These results are summarized in Tables 2-4.

Overall, we still conclude that GRU is capable to reliably mitigate the trade-off between removal and retention, typically showing improvements for all the metrics that align with each goal. Note that, we also observe that in some situations, the enhancements in preserving overall model performance occur at the expense of decreased strength of removal, particularly for those results on WMDP and MUSE. Fortunately, this scenario occurs only for certain specific metrics, and the decrease in the efficacy of removal appears to be negligible when compared to the substantial improvements in retention. Therefore, we still consider our GRU as an effective solution to mitigate the trade-off and enhance the overall unlearning efficacy.

*Table 2.* Full experimental results on the TOFU benchmarks: Evaluating 5% and 10% unlearning setups across different backbones and baseline methods. The results are presented in two adjacent rows for each method, one row (original baseline method name) showing the original results and the other (w/ GRU) displaying the results combined with GRU. The superior results between configurations with and without GRU for each baseline method are highlighted in **bold**.

| Method | Phi-1.5 | | | | LLaMA2-7B | | | |
| | 5% | | 10% | | 5% | | 10% | |
| | FQ↑ | MU↑ | FQ↑ | MU↑ | FQ↑ | MU↑ | FQ↑ | MU↑ |
|---|---|---|---|---|---|---|---|---|
| Original | -28.8476 | 0.5200 | -40.5243 | 0.5200 | -32.1330 | 0.6332 | -48.5895 | 0.6332 |
| Retrain | 0.0000 | 0.5250 | 0.0000 | 0.5320 | 0.0000 | 0.6006 | 0.0000 | 0.6137 |
| GA | -28.0555 | 0.0000 | -28.5669 | 0.0000 | -16.9281 | 0.0000 | -14.3716 | 0.0000 |
| w/ GRU | **-5.1004** | **0.3587** | **-11.3678** | **0.2482** | **-3.5161** | **0.5190** | **-12.1912** | **0.3108** |
| WGA | -12.4230 | 0.0284 | -29.8615 | 0.0063 | -7.7503 | 0.4447 | -28.5669 | 0.1154 |
| w/ GRU | **-1.9514** | **0.4431** | **-6.6882** | **0.4184** | **-5.1004** | **0.5698** | **-19.7868** | **0.5107** |
| NPO | -11.9082 | 0.3565 | -15.7638 | 0.3267 | -10.9105 | 0.4919 | -8.7037 | 0.2876 |
| w/ GRU | **-0.9326** | **0.3935** | **-3.1620** | **0.3714** | **-9.9550** | **0.5408** | **-2.5106** | **0.4570** |
| GD | -6.5526 | 0.4061 | -50.2968 | 0.2999 | -13.4847 | 0.5549 | -13.9215 | 0.3930 |
| w/ GRU | **-5.8059** | **0.4138** | **-13.4785** | **0.4096** | **-12.4230** | **0.5637** | **-11.7760** | **0.5407** |
| NPO+KL | -11.9082 | 0.3634 | -17.2193 | 0.3444 | -10.4275 | 0.5094 | -9.4304 | 0.3109 |
| w/ GRU | **-0.0360** | **0.3833** | **-3.1620** | **0.3654** | **-10.4275** | **0.5585** | **-2.1101** | **0.4480** |
| NPO+GD | -12.4230 | 0.4002 | -19.7868 | 0.4026 | -11.9082 | 0.5256 | -12.6133 | 0.4750 |
| w/ GRU | **-9.4931** | **0.4514** | **-7.6651** | **0.4122** | **-8.1703** | **0.5673** | **-2.9381** | **0.5000** |
| SimNPO+GD | **-12.9485** | 0.4428 | -26.6801 | 0.4523 | -9.0417 | 0.5073 | -9.8040 | 0.5527 |
| w/ GRU | **-12.9485** | **0.4862** | **-25.4588** | **0.4934** | **-9.0417** | **0.5516** | **-9.4304** | **0.6168** |

*Table 3.* Detailed experimental results on the WMDP benchmarks with QA accuracies evaluated on Bio unlearning and MMLU using the ZEPHYR-7B-BETA backbone. The results are presented in two adjacent rows for each method, one row (original baseline method name) showing the original results and the other (w/ GRU) displaying the results combined with GRU. The superior results between configurations with and without GRU for each baseline method are highlighted in **bold**.

| Method | Unlearning | | Retention |
|---|---|---|---|
| | Bio ↓ | Cyber ↓ | MMLU ↑ |
| Original | 0.6371 | 0.4383 | 0.5814 |
| GA | **0.2474** | **0.2431** | 0.2465 |
| w/ GRU | **0.2474** | 0.2446 | **0.2852** |
| WGA | 0.2476 | 0.2647 | 0.2454 |
| w/ GRU | **0.2474** | **0.2587** | **0.2604** |
| GD | **0.2474** | **0.2441** | 0.2589 |
| w/ GRU | **0.2474** | 0.2511 | **0.2995** |
| NPO | 0.2655 | **0.2793** | 0.3033 |
| w/ GRU | **0.2561** | **0.2793** | **0.3704** |
| NPO+GD | 0.2710 | **0.3493** | 0.4724 |
| w/ GRU | **0.2639** | 0.3524 | **0.5033** |

*Table 4.* Detailed experimental results on the MUSE benchmarks with KnowMem, assessing the efficacy of removal and retention on targeted and non-targeted data, respectively. The results are presented in two adjacent rows for each method, one row (original baseline method name) showing the original results and the other (w/ GRU) displaying the results combined with GRU. The superior results between configurations with and without GRU for each baseline method are highlighted in **bold**.

| Method | VerbMem ↓ | KnowMem-U ↓ | KnowMem-R ↑ |
|---|---|---|---|
| Original | 99.7016 | 45.8791 | 69.4009 |
| Retrain | 13.8896 | 30.1380 | 69.0496 |
| GA | **0.0000** | **0.0000** | 0.0000 |
| w/ GRU | **0.0000** | **0.0000** | **6.7006** |
| WGA | 0.2284 | **0.0000** | 0.0000 |
| w/ GRU | **0.0198** | **0.0000** | **18.4555** |
| GD | **0.0000** | **0.0000** | 8.6971 |
| w/ GRU | **0.0000** | **0.0000** | **9.8586** |
| NPO | **0.0000** | **0.0000** | 0.0000 |
| w/ GRU | **0.0000** | **0.0000** | **9.5913** |
| NPO+GD | **0.0000** | **0.0000** | 23.5565 |
| w/ GRU | **0.0000** | **0.0000** | **30.7492** |

## C. Comparison with RMU on WMDP

For completeness, we compare our approach with RMU, the method proposed alongside WMDP. Due to sensitivity issues noted in the official implementation, we set the hyperparameter $\alpha$ to 100 (instead of the default 1200) to ensure stable optimization. Table 5 reports the results for both RMU and its combination with GRU.

*Table 5.* Comparison of RMU and RMU with GRU on the WMDP benchmark (Bio and Cyber: accuracy $\downarrow$; MMLU: accuracy $\uparrow$).

| Method | Bio $\downarrow$ | Cyber $\downarrow$ | MMLU $\uparrow$ |
|---|---|---|---|
| RMU | 0.26 | 0.31 | 0.41 |
| w/ GRU | 0.26 | 0.28 | 0.44 |

As shown, GRU consistently improves both unlearning and retention over the RMU baseline.

## D. Hyper-parameter Analyses

In addition to our main results, we further discuss about our hyper-parameter configurations as well as conduct additional analyses on hyper-parameter sensitivity.

### D.1. Hyper-parameter Configurations

We employ grid search on validation data to select proper hyper-parameters for GRU and TRU. For GRU, the candidate values for $\gamma$ include $\{0.01, 0.05, 0.1, 0.2, 0.3, 0.4, 0.5, 0.6, 0.7, 0.8, 0.9, 0.99\}$, and that for $\tau$ are $\{0.001, 0.005, 0.01, 0.1, 1.0, 10, 100\}$. For TRU, we select `stg` from the space $\{0.50, 0.60, 0.65, 0.70, 0.75, 0.80, 0.85\}$ while fix $\text{lr} = 1e^{-4}$. We further summarize their detailed configurations across different setups as follows.

**GRU.** For TOFU with Phi-1.5, we default to set $\tau = 0.001$, while adjusting $\tau = 0.01$ for the 5% setup and without using gradient clipping for GD. For TOFU with Llama-2-7B, we do not use gradient clipping for GA, GD, NPO, and SimNPO under the 5% setup, while setting $\tau = 1.0$ for all other methods. In the 10% setup, $\tau = 1.0$ for GA, WGA, GD, SimNPO, and NPO+GD; $\tau = 0.5$ for NPO; $\tau = 0.1$ for NPO+KL. For WMDP, $\tau = 1.0$ for GA; $\tau = 0.01$ for NPO and NPO+GD; $\tau = 0.001$ for GD and WGA. For MUSE, $\tau = 1.0$ for GA, GD and WGA; $\tau = 100$ for NPO and NPO+GD.

For TOFU, we by default set $\gamma = 0.8$, while setting $\gamma = 0.05$ with Llama-2-7B and and $\gamma = 0.1$ with Phi. Also, we set $\gamma = 0.5$ for SimNPO. Moreover, we set $\gamma = 0.8$ for MUSE and $\gamma = 0.99$ for WMDP.

**TRU.** With the backbone of Phi-1.5, we set `stg` = 0.7 under the 5% setup and `stg` = 0.75 under the 10% setup. Also, with the backbone of Llama-2-7B, we set `stg` = 0.65 under the 5% setup and `stg` = 0.85 under the 10% setup.

### D.2. Sensitivity Analyses

As a case study, we conduct sensitivity analyses on TOFU with Llama-2-7B as the backbone, under the 5% unlearning setup.

**Gradient Clipping.** We first present the results across various values of $\tau$, summarized in Table 6. The results show that, across different baselines, the effects of altering $\tau$ have a smooth control over the performance metrics of FQ and MU. This observation indicates that our GRU exhibits robustness with respective to different choices of $\tau$.

**Exponential Moving Average.** We further display the results across different $\gamma$ in Table 7. As with the gradient clipping, we observe a smooth control on the overall efficacy of unlearning, further indicating that our GRU method demonstrates robustness against variations in its two hyper-parameters.

## E. Ablation Studies and Other Analyses

We provide more analyses to further show the respective effects of different components involved in our algorithm design.

**Ablation Studies.** Previous works, such as (Zhang et al., 2024), also use gradient clipping (GC) to improve the overall efficacy of unlearning, raising us to ask if our rectification mechanism plays a key role to mitigate the trade-off between removal and retention. In Table 8, we conduct ablation studies on TOFU using Llama-2-7B as the backbone, focusing on the

*Table 6.* Hyper-parameter turning of $\tau$ on TOFU with Llama-2-7B, under the 5% unlearning setup.

| Method | Metric | $\tau$ | | | | | | | | |
|---|---|---|---|---|---|---|---|---|---|---|
| | | 0.001 | 0.01 | 0.1 | 1.0 | 2.0 | 3.0 | 10 | 100 | N/A |
| GA w/ GRU | FQ ↑ | -29.6514 | -15.7370 | -7.7503 | -8.6008 | -4.7631 | -4.4360 | -2.9534 | -3.2299 | -3.5161 |
| | MU ↑ | 0.6326 | 0.5824 | 0.5761 | 0.5810 | 0.5684 | 0.5550 | 0.5121 | 0.5149 | 0.5190 |
| NPO w/ GRU | FQ ↑ | -27.2750 | -18.7967 | -12.9485 | -9.0417 | -9.4931 | -9.9550 | -9.9550 | -10.4275 | -9.9550 |
| | MU ↑ | 0.6268 | 0.5796 | 0.5574 | 0.5220 | 0.5519 | 0.5318 | 0.5312 | 0.5373 | 0.5408 |
| GD w/ GRU | FQ ↑ | -27.2750 | -18.7967 | 14.0316 | -15.1577 | -14.5893 | -14.5893 | -15.7370 | -12.4230 | -12.4230 |
| | MU ↑ | 0.6200 | 0.5395 | 0.5442 | 0.5434 | 0.5467 | 0.5467 | 0.5484 | 0.5637 | 0.5637 |

*Table 7.* Hyper-parameter turning of $\gamma$ on TOFU with Llama-2-7B, under the 5% unlearning setup. The notation "–" indicates that the associated result is same to those without GRU.

| Method | Metric | $\gamma$ | | | | | | | | | | | | |
|---|---|---|---|---|---|---|---|---|---|---|---|---|---|---|
| | | 0.01 | 0.02 | 0.05 | 0.1 | 0.2 | 0.3 | 0.4 | 0.5 | 0.6 | 0.8 | 0.9 | 0.99 | N/A |
| GA w/ GRU | FQ ↑ | -6.9414 | -5.8059 | -6.9414 | -7.3407 | -8.1703 | -7.3407 | -6.5526 | -7.7503 | -6.5526 | -8.6008 | -6.5526 | -8.6008 | -8.1703 |
| | MU ↑ | 0.4528 | 0.4418 | 0.4715 | 0.4994 | 0.5247 | 0.5504 | 0.5601 | 0.5677 | 0.5739 | 0.5810 | 0.5827 | 0.5847 | 0.5829 |
| NPO w/ GRU | FQ ↑ | -11.9082 | -11.4040 | -9.9550 | -9.9550 | -9.4931 | -8.6008 | -8.1703 | -9.0417 | -9.0417 | -9.0417 | -10.4275 | -10.9105 | -9.0417 |
| | MU ↑ | 0.4736 | 0.4773 | 0.4865 | 0.4952 | 0.5108 | 0.5144 | 0.5159 | 0.5178 | 0.5222 | 0.5220 | 0.5425 | 0.5358 | 0.5656 |
| GD w/ GRU | FQ ↑ | -12.9485 | -12.4230 | -12.4230 | -14.0316 | -12.9485 | -11.9082 | -14.0316 | -14.0316 | -13.4847 | – | – | – | – |
| | MU ↑ | 0.5642 | 0.5640 | 0.5637 | 0.5615 | 0.5604 | 0.5585 | 0.5586 | 0.5592 | 0.5582 | – | – | – | – |

5% unlearning setup. We compare three scenarios: the original unlearning method without GRU (w/o GRU), the original method enhanced with GC (w/ GC), and the unlearning method that incorporates GRU (w/ GRU). As evident from the results, GRU demonstrates superior scores in terms of both FQ and MU, showing its efficacy in mitigating the trade-off between removal and retention.

*Table 8.* Ablation studies on TOFU with Llama-2-7B, under the 5% unlearning setup.

| Method | Component | FQ ↑ | MU ↑ |
|---|---|---|---|
| GA | w/o GRU | -16.9281 | 0.0000 |
| | w/ GC | -20.7646 | 0.0000 |
| | w/ GRU | **-3.2299** | **0.5149** |
| NPO | w/o GRU | -10.9105 | 0.4919 |
| | w/ GC | -10.9105 | 0.4970 |
| | w/ GRU | **-10.4275** | **0.5373** |

**Visualization of Rectification.** We further examine the angles between $g_{\mathrm{u}}^{(t)}$ and $g_{\mathrm{r}}^{(t)}$, along with those results after being rectified via GRU. We monitor the dynamics of these angles throughout the unlearning processes for various baseline methods, as well as the changes after applying GRU for rectification. As a case study, Figure 7 shows these results on TOFU 5% unlearning with Llama-2-7B as the backbone. Without the use of GRU, the cosine similarity between $g_{\mathrm{u}}^{(t)}$ and $g_{\mathrm{r}}^{(t)}$ keeps negative throughout unlearning, suggesting potential adverse effects on the overall performance of the model. In comparison, within the unlearning dynamics facilitated by GRU, it is observed that although the angles initially continue to be negative (dotted lines), our rigorous method of gradient rectification will adjust the resulting cosine similarity to exactly 0. This adjustment ensures that the gradient direction associated with unlearning is completely orthogonal to that of retention, thereby effectively maintaining the overall model performance.

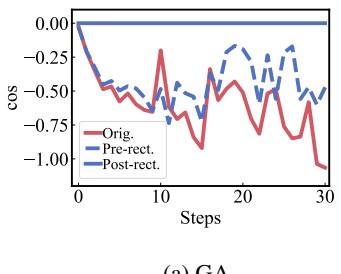

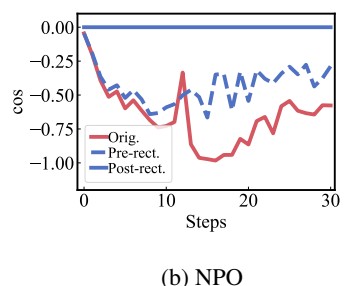

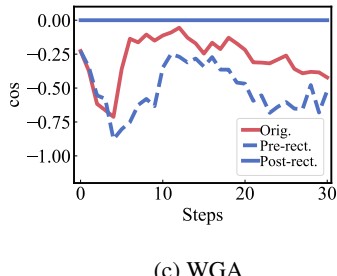

(a) GA         (b) NPO         (c) WGA

*Figure 7.* Angles between $g_u^{(t)}$ and $g_r^{(t)}$ (labeled 'Orig.' for the original baseline and 'Pre-rect.' for the GRU-enhanced baseline) and between $\tilde{g}_u^{(t)}$ and $g_r^{(t)}$ (labeled 'Post-rect.' for the GRU-enhanced baseline) across various unlearning steps. We evaluate the baseline methods NPO, GA, and WGA in a 5% TOFU setup using the Llama-2-7B model.

## F. Aligning Unlearning with Retention: A Practically Meaningful Evaluation

Recalling the dual goals of LLM unlearning, a meaningful evaluation requires not only improvements in the forget set, but also that the retention utility of the model remains well aligned with its original capabilities. A substantial decline in utility performance would render the resulting model ineffective, making the process of LLM unlearning itself meaningless. This concern, often overlooked in recent unlearning studies, can be addressed by **Unlearning with Control (UWC)** (Wang et al., 2024a), as discussed in Section 2.2. UWC provides a post-unlearning calibration framework that restores retention performance by interpolating the model parameters before and after unlearning via a tunable parameter $\alpha$. Leveraging UWC calibration, we systematically evaluate our approach under explicit retention thresholds (e.g., 85%, 90%, and 95%) to investigate whether strong unlearning performance can be achieved without sacrificing essential retention utility, thus aligning unlearning objectives with practical deployment needs. Given the similarity of our findings across multiple benchmarks, we focus here on representative results obtained from the challenging Phi setup on TOFU. Specifically, we present GA and NPO as baseline methods to illustrate the flexibility of UWC calibration and highlight the effectiveness of our GRU approach in attaining superior unlearning results while rigorously maintaining retention utility. The results in Tables 9 and 10 clearly demonstrate that, under each retention constraint, incorporating GRU consistently yields substantial improvements in forget quality (FQ) compared to the calibrated baselines, while perfectly maintaining the prescribed model utility (MU). This pattern holds for both GA and NPO methods, across all retention targets and unlearning setups. These findings validate the practical value of our approach: by leveraging UWC calibration in combination with GRU, practitioners can achieve strong, controllable unlearning effects without sacrificing the essential utility of large language models, thereby ensuring that unlearning objectives remain aligned with real-world deployment requirements.

*Table 9.* GA on the TOFU Phi-1.5 setup with UWC calibration. We report FQ (forget quality, ↑) and MU (model utility, ↑) for 5% and 10% unlearning under three retention targets (85%, 90%, 95%). Each retention target is shown in two adjacent rows: the first row gives the calibrated GA result, and the second (w/ GRU) shows the result after incorporating GRU. The better score within each GA–GRU pair is in **bold**.

| Method | 5% Unlearning | | 10% Unlearning | |
|---|---|---|---|---|
| | FQ↑ | MU↑ | FQ↑ | MU↑ |
| Original | -28.8 | 0.52 | -40.5 | 0.52 |
| GA (85%) | -22.0 | 0.44 | -35.3 | 0.44 |
| w/ GRU (85%) | **-8.6** | 0.44 | **-20.8** | 0.44 |
| GA (90%) | -28.1 | 0.47 | -36.8 | 0.47 |
| w/ GRU (90%) | **-15.2** | 0.47 | **-28.8** | 0.47 |
| GA (95%) | -28.1 | 0.49 | -39.8 | 0.49 |
| w/ GRU (95%) | **-18.8** | 0.49 | **-33.9** | 0.49 |

*Table 10.* NPO on the TOFU Phi-1.5 setup with UWC calibration. Layout and notation follow Table 9.

| Method | 5% Unlearning | | 10% Unlearning | |
|---|---|---|---|---|
| | FQ↑ | MU↑ | FQ↑ | MU↑ |
| Original | -28.8 | 0.52 | -40.5 | 0.52 |
| NPO (85%) | -15.7 | 0.44 | -31.9 | 0.44 |
| w/ GRU (85%) | **-9.5** | 0.44 | **-15.2** | 0.44 |
| NPO (90%) | -20.1 | 0.47 | -35.3 | 0.47 |
| w/ GRU (90%) | **-12.9** | 0.47 | **-18.2** | 0.47 |
| NPO (95%) | -25.0 | 0.49 | -38.2 | 0.49 |
| w/ GRU (95%) | **-14.0** | 0.49 | **-20.8** | 0.49 |

# G. Comparison with Gradient Direction Rectification (GDR)

Closely related is Gradient Direction Rectification (GDR) (Lin et al., 2024), as it similarly employs gradient projection to resolve conflicts between forgetting and retention objectives. However, GDR relies on caching gradients across epochs, resulting in substantial memory overhead that limits its scalability for large language models, where parameter sizes are massive and training typically involves only a few epochs. In contrast, GRU dynamically estimates retention gradients using an exponential moving average, greatly reducing memory cost and enabling practical unlearning at scale. Furthermore, while GDR merges retention gradients directly into parameter updates, potentially increasing the risk of overfitting to the retention set, our GRU approach leverages retention gradients solely as constraints for rectification. Finally, TRU extension addresses the challenge of biased retain data, a scenario unique to LLM unlearning and not considered in GDR.

