# OpenReview forum: "GRU: Mitigating the Trade-off between Unlearning and Retention for LLMs"
_ICML.cc/2025/Conference — ICML 2025 poster_

### Official Review · Reviewer_qBZD · 2025-02-25

**Overall Recommendation:** 3

**Summary:**

This paper is concerned with the problem of large language model unlearning, which is the process of removing a specific piece of information from a pre-trained language model. The authors note that unlearning usually comes with the cost of harming the performance of the model on other tasks. To mitigate this, they proposed `Gradient Rectified Unlearning` (GRU). The key idea of GRU is to project the gradient of the unlearning loss onto a perpendicularly aligned subspace of the gradient of the retention loss. The authors show both theoretical and empirical results to demonstrate the effectiveness of GRU. In addition, they also consider a setting where we only have unlearning data but no retention data, and propose a method called `Task Vector Rectified Unlearning` (TRU) based on an idea similar to GRU. The authors empirically show that TRU can achieve better performance over baseline methods in this setting.

## Update after rebuttal

I am glad to see the authors have incorporated RMU into their experiment design.
Overall, I am OK with this paper given that you will incorporate these changes into the final version and have raised my score to 3. However, I won't champion it since the changes in the experiment design are kinda significant.

**Claims And Evidence:**

I find the theoretical claims convincing, while the empirical results are not very strong, mainly because of the choice of baseline methods. See the detailed comments in `Experimental Designs Or Analyses`.

**Essential References Not Discussed:**

See the comments in `Experimental Designs Or Analyses`.

**Experimental Designs Or Analyses:**

A major concern about the empirical results is the choice of baseline methods. The authors consider a set of baselines in the main experiments, including GA, GD, NPO and WGA. I think the choice of baselines is not very strong. Among these methods, NPO is probably the best one, but it is still not considered as state-of-the-art. Given the authors are already using the WMDP benchmark, I would suggest them to consider RMU, which is proposed in the same paper. From my own research experience, RMU performs better than NPO in the trade-off between unlearning and retention significantly. Without considering RMU, the empirical improvements of GRU and TRU over the baselines are not very convincing to me.

**Methods And Evaluation Criteria:**

The proposed methods and evaluation criteria make sense.

**Other Comments Or Suggestions:**

Line 254: `datat` should be `data`.

Line 255: `eeach` should be `each`.

Line 267: $\mathbf T_s$ is not defined.

**Other Strengths And Weaknesses:**

The paper is well-written and easy to follow, and I think the idea of GRU is neat.

Figure 3 is a bit confusing. Each algorithm's performance with and without GRU is shown as a segment in the figure. But it is unclear to me which endpoint of the segment is with GRU, i.e., whether GRU makes the performance better or worse. I can only guess from the caption that GRU improves the performance, but I think it would be necessary to make it clearer in the figure or at least in the caption.

**Questions For Authors:**

What is the rationale behind the choice of baseline methods?

**Relation To Broader Scientific Literature:**

This paper's idea of projecting the gradient of the unlearning loss onto a perpendicularly aligned subspace of the gradient of the retention loss is potentially useful for many algorithms in the context of unlearning, as the authors demonstrated in the paper. But it remains unclear if this improvement applies to state-of-the-art methods like RMU.

**Theoretical Claims:**

I read the statements in the paper and skimmed through the proofs in the appendix. I did not check the proofs in detail, but there is no reason to doubt the correctness of the theoretical claims.

---

> ### Author Rebuttal · Authors · 2025-03-31
>
> Thank you sincerely for your constructive comments and for helping us identify typos. We hope that the feedback provided below will address your concerns.
>
> > Q1. A major concern about the empirical results is the choice of baseline methods. Given that the authors are already using the WMDP benchmark, I would suggest them to consider RMU, which is proposed in the same paper. From my own research experience, RMU performs better than NPO in the trade-off between unlearning and retention significantly. Without considering RMU, the empirical improvements of GRU and TRU over the baselines are not very convincing to me.
>
> **A1**. We aim to use a general set of representative methods for benchmarking. Regrettably, RMU, possibly due to its complexity to be implemented, is not widely adopted in earlier studies, such as [1,2] (though both cited WMDP). Although some recent papers incorporated RMU, their evaluations are either limited to WMDP [3], or they reveal its sensitivity to hyperparameters [4].
>
> However, we completely agree that RMU represents a state-of-the-art within WMDP, which merits our particular focus. We present the results using RMU on WMDP, along with its version with GRU. As observed, GRU indeed enhance performance for both retention and unlearning, showing our GRU is general to mitigate the trade-off between unlearning and retention.
> Method|WMDP Bio↓|WMDP Cyber↓|WMDP MMLU↑|
> |:-:|:-:|:-:|:-:|
> |RMU|0.26|0.31|0.41|
> |w/ GRU|0.26|0.28|0.44|
>
>
> We adjust $\alpha$ from the default 1200 to 100 in the RMU open-sourced code, after finding that the original settings caused the retain term to overly dominate, hindering model updates. Similar sensitivity issues, such as precision setups, have been reported by others in the WMDP GitHub repository. We will study these issues in the future. However, we continue to see the RMU as a promising approach, particularly when viewing through the lens of local knowledge perturbation. We plan to delve deeper into the RMU concept and explore more specific verisons of GRU for RMU (e.g., constraints for masking) in our future work.
>
> [1] Negative Preference Optimization: From Catastrophic Collapse to Effective Unlearning.
>
> [2] MUSE: Machine Unlearning Six-Way Evaluation for Language Models.
>
> [3] Simplicity Prevails: Rethinking Negative Preference Optimization for LLM Unlearning.
>
> [4] Rethinking LLM Unlearning Objectives: A Gradient Perspective and Go Beyond.
>
>
> > Q2. Figure 3 is a bit confusing. Each algorithm's performance with and without GRU is shown as a segment in the figure. But it is unclear to me which endpoint of the segment is with GRU, i.e., whether GRU makes the performance better or worse. I can only guess from the caption that GRU improves the performance, but I think it would be necessary to make it clearer in the figure or at least in the caption.
>
>
> **A2**. Apologies for any confusion caused by the figure annotations. As detailed in the figure caption, each pair of scores represents metric values (either FQ or MU) before and after applying the GRU enhancement. For instance, taking Figure 3(a) as an example, the pair of scores (-16.93, -3.52) represents the FQ scores for the GA method, where -16.93 is the score without the use of GRU and -3.52 is the score with GRU. The visual representation using an upward growing grid between these scores emphasizes the improvements achieved by incorporating GRU. We will clarify the meaning of our figures and provide more explicit explanations in our revision to ensure clear understanding.

---

> > ### Comment · Reviewer_qBZD · 2025-04-01
> >
> > Thank you for your reply.
> >
> > - I am glad to see you have incorporated RMU into your experiment design.
> > - To say a bit more about my confusion with the figures, note that you are stating two results - with and without GRU - where it is not necessary that GRU provides an improvement. Therefore, it's better to display the results well.
> >
> > Overall, I am OK with this paper given that you will incorporate these changes into the final version and have raised my score to 3. However, I won't champion it since the changes in the experiment design are kinda significant.

---

> > > ### Author Response · Authors · 2025-04-02
> > >
> > > Sincere thanks for your great support and the raised score, which mean a great deal to us! We will absolutely include additional results and better visualization in our revision and consult the authors of RMU to address the hyper-parameter configuration issue. We are committed to conducting further experiments on RMU. Thank you once again for your comments and support!

---

### Official Review · Reviewer_iHZj · 2025-03-12

**Overall Recommendation:** 3

**Summary:**

This paper addresses the problem of Machine Unlearning in Large Language Models (LLMs). To balance performance between the retain set and the forget set, the unlearning update is adjusted to avoid harming the performance on the retain set during the unlearning process. Starting from a fundamental optimization problem designed to implement this idea, the paper presents its closed-form solution, along with pseudocode and the theoretical foundations supporting the approach. The proposed technique is evaluated on WMDP, MUSE, and TOFU benchmarks, and it demonstrates successful performance improvements when combined with the loss functions of GA, GD, and NPO.

**Claims And Evidence:**

The claims of the proposed method are clearly presented. Although there are some weaknesses, the paper is overall well-written and solid. Below, I will provide a more detailed discussion on these points.

**Essential References Not Discussed:**

There was nothing in particular.

**Experimental Designs Or Analyses:**

Although hyperparameter settings, such as learning rate (LR), are specified in the paper, the process by which they were determined is not described. That said, the paper does explain how the hyperparameters for the proposed GRU method were selected. However, some questions remain. In particular, how was the validation data chosen? Since there is no widely established standard for selecting validation data in unlearning settings, I believe the paper should provide a more detailed explanation on this aspect.

From a metric perspective, the evaluation feels somewhat insufficient. For instance, only WMDP-Cyber and MUSE-KnowMem results are reported, but VerbMem results are missing. I think that evaluating the proposed method on only a subset of metrics provided by the benchmarks does not fully demonstrate its effectiveness. A more comprehensive evaluation, covering all relevant metrics such as VerbMem, would strengthen the paper's claims.

**Methods And Evaluation Criteria:**

The authors use three benchmarks, which seems reasonable. For the evaluation metrics, they adopt those proposed in each benchmark, which is also appropriate. Regarding the performance on the forget set, they show how much improvement is achieved when combined with existing loss functions, which appears to be a reasonable approach.

However, I have some concerns about the evaluation on the retain set. Specifically, I question whether the current way of handling the retain set is sufficient. In my view, unless the utility performance on the retain set (e.g., MU or similar metrics) remains at 90–95% of the original model's performance, the improvements on the forget set may have limited practical meaning because the model's overall performance as a language model would already be compromised.

Therefore, as discussed in the Task Arithmetic paper, I believe it is more appropriate to constrain the retain set performance to be at least 95% of the original model and then examine how much forgetting can be achieved under that constraint. Including experimental results from this perspective would significantly strengthen the paper.

**Other Comments Or Suggestions:**

N/A

**Other Strengths And Weaknesses:**

The paper was overall well-written and easy to understand. The logical flow of arguments supporting the claims was excellent.

**Questions For Authors:**

I recognize that my perspective on unlearning evaluation differs from that of the authors. While it may not be entirely fair to insist that my view is the only correct one, I still struggle to understand the significance of forget set performance when the model's overall capability as a language model is already degraded. From my perspective, if the utility performance drops significantly, any gain on the forget set becomes less meaningful.

It would be very helpful if the authors could either provide a counter-argument to this view or present experimental results where the utility performance is controlled to remain at 90–95% of the original model, so that the trade-off between the retain and forget sets can be properly evaluated.

For reference, my overall recommendation is borderline. However, since there is no borderline option this time, I selected weak reject. I plan to revisit and potentially update my score after reading the author response letter.

**Relation To Broader Scientific Literature:**

The paper is well-connected to the existing literature, as it addresses the problem of severe performance degradation on the retain set caused by previous unlearning methods.

**Theoretical Claims:**

This paper makes a theoretical claim, which appears reasonably sound upon skimming. However, I did not rigorously verify the theoretical development by working through the derivations in detail.

---

> ### Author Rebuttal · Authors · 2025-03-31
>
> Sincere thanks for your constructive comments, and we hope the following feedbacks can address your concerns.
>
> > Q1. Specifically, I question whether the current way of handling the retain set is sufficient. In my view, unless the utility performance on the retain set (e.g., MU or similar metrics) remains at 90–95% of the original model's performance, the improvements on the forget set may have limited practical meaning because the model's overall performance as a language model would already be compromised.
>
> **A1**. We totally agree with your opinion. A large decline in utility performance would indeed render the resulting model ineffective, at which point the process of LLM unlearning also becomes meaningless. On the other side, carefully tuning the hyperparameters to achieve the goal of preserving 85, 90, and 95% of the original model performance can be tedious. Fortunately, as mentioned in Section 2.2, the UWC method offers a post-unlearning strategy that enables calibrating model performance via model mixing, alleviating the challenges associated with maintaining utility. Due to the relatively high cost of UWC, here we take GA and NPO under the challenging Phi setup as two examples to show UWC's flexibility and GRU's effectiveness. The improvements achieved by GRU are more pronounced. We will add more results in our revision. Many thanks for your suggestion.
>
> w/ UWC |FQ 5%↑|MU 5%↑|FQ 10%↑|MU 10%↑|
> |:-:|:-:|:-:|:-:|:-:|
> |Original|-28.8|0.52|-40.5|0.52|
> |GA (85%)|-22.0|0.44|-35.3|0.44|
> |w/ GRU (85%)|-8.6|0.44|-20.8|0.44|
> |GA (90%)|-28.1|0.47|-36.8|0.47|
> |w/ GRU (90%)|-15.2|0.47|-28.8|0.47|
> |GA (95%)|-28.1|0.49|-39.8|0.49|
> |w/ GRU (95%)|-18.8|0.49|-33.9|0.49|
>
> w/ UWC |FQ 5%↑|MU 5%↑|FQ 10%↑|MU 10%↑|
> |:-:|:-:|:-:|:-:|:-:|
> |Original|-28.8|0.52|-40.5|0.52|
> |NPO (85%)|-15.7|0.44|-31.9|0.44|
> |w/ GRU (85%)|-9.5|0.44|-15.2|0.44|
> |NPO (90%)|-20.1|0.47|-35.3|0.47|
> |w/ GRU (90%)|-12.9|0.47|-18.2|0.47|
> |NPO (95%)|-25.0|0.49|-38.2|0.49|
> |w/ GRU (95%)|-14.0|0.49|-20.8|0.49|
>
> > Q2. In particular, how was the validation data chosen? Since there is no widely established standard for selecting validation data in unlearning settings, I believe the paper should provide a more detailed explanation on this aspect.
>
> **A2**. Many thanks for your comments. This perspective has indeed been overlooked by the majority of the community, and we appreciate the opportunity to detail our implementation below.
>
> For **unlearn data**, the datasets used for validation are exactly those used for unlearning **in the cases of TOFU and MUSE**, as they focus on privacy and copyright removal and do not require generalization. However, **for WMDP**, which aims to unlearn the entire concepts of harmful subjects, generalization is crucial. Hence, we separate a small set of test data (comprising 200 randomly selected samples) for evaluation, with respect to both the bio and cyber setups. For **retain data**, we separate 200 (20 for MUSE Book due to its small size of retain data) randomly selected samples from the original retain datasets used for unlearning, with respect to all three benchmarks. It ensures that we have a distinct set of data to verify retention compared to test data. We will add a detailed discussion in our revision.
>
> > Q3. Only WMDP-Cyber and MUSE-KnowMem results are reported, but VerbMem results are missing.
>
> **A3**. The results for VerbMem are in Appendix B. We did not include these in the main text as their values and ranks are quite similar to KnowMem-U, offering limited new insights. We apologize for any confusion caused and will clarify this choice in our revision.

---

> > ### Comment · Reviewer_iHZj · 2025-04-02
> >
> > I appreciate that the authors agreed with my perspective. However, they only provided results showing controlled performance on the retain set for the TOFU dataset. Without knowing how the method performs on MUSE and WMDP in the same setting, it is difficult for me to recommend acceptance of the paper. Unfortunately, I will maintain my initial rating.

---

> > > ### Author Response · Authors · 2025-04-03
> > >
> > > Many thanks for your further comments! In our initial responses, we aimed to show the possibility to control retention performance post-unlearning based on the well-established UWC calibration framework. We are more than happy to provide more aligned results (85, 90, and 95%) across other benchmarks (WMDP and MUSE). We present the results below, which further verify our effectiveness in mitigating the trade-off.
> > >
> > > Method|WMDP Bio↓|WMDP Cyber↓|WMDP MMLU↑|
> > > |:-:|:-:|:-:|:-:|
> > > |GA (85%)|0.44|0.40|0.49|
> > > |w/ GRU (85%)|0.37|0.35|0.49|
> > > |NPO (85%)|0.34|0.39|0.49|
> > > |w/ GRU (85%)|0.25|0.36|0.49|
> > > |GD (85%)|0.31|0.37|0.49|
> > > |w/ GRU (85%)|0.29|0.35|0.49|
> > > |GA (90%)|0.53|0.41|0.52|
> > > |w/ GRU (90%)|0.46|0.39|0.52|
> > > |NPO (90%)|0.45|0.40|0.52|
> > > |w/ GRU (90%)|0.25|0.39|0.52|
> > > |GD (90%)|0.37|0.40|0.52|
> > > |w/ GRU (90%)|0.35|0.37|0.52|
> > > |GA (95%)|0.60|0.42|0.55|
> > > |w/ GRU (95%)|0.55|0.41|0.55|
> > > |NPO (95%)|0.54|0.42|0.55|
> > > |w/ GRU (95%)|0.26|0.41|0.55|
> > > |GD (95%)|0.56|0.43|0.55|
> > > |w/ GRU (95%)|0.44|0.41|0.55|
> > >
> > >
> > > Method |MUSE VerbMem ↓|MUSE KnowMem-U ↓|MUSE KnowMem-R ↑|MUSE PrivLeak → 0|
> > > |:-:|:-:|:-:|:-:|:-:|
> > > |GA (85%)    |98|41|59|-56|
> > > |w/ GRU (85%) |7|32|59|-39|
> > > |NPO (85%)|55|28|59|-50|
> > > |w/ GRU (85%)|0|21|59|-11|
> > > |GD (85%)   |98|42|59|-53|
> > > |w/ GRU (85%)|84|41|59|-51|
> > > |GA (90%)    |100|45|62|-52|
> > > |w/ GRU (90%) |97|42|62|-52|
> > > |NPO (90%)|95|39|62|-53|
> > > |w/ GRU (90%)|0|22|62|-15|
> > > |GD (90%)   |98|44|62|-54|
> > > |w/ GRU (90%)|96|43|62|-52|
> > > |GA (95%)    |100|47|66|-54|
> > > |w/ GRU (95%) |98|45|66|-53|
> > > |NPO (95%)|91|41|66|-53|
> > > |w/ GRU (95%)|2|25|66|-26|
> > > |GD (95%)   |100|47|66|-54|
> > > |w/ GRU (95%)|98|46|66|-53|
> > >
> > >
> > > We fully agree that the aligned performance facilitates a fairer and clearer comparison of unlearning efficacy. We will certainly add more results and the related discussions in our revision.
> > >
> > > We hope our new results can address your concerns. Please let us know if you need any further information or if there are additional points you would like to discuss with us. We are excited for further discussions with you!

---

### Official Review · Reviewer_z9sW · 2025-03-14

**Overall Recommendation:** 3

**Summary:**

The paper introduces Gradient Rectified Unlearning, an unlearning framework that constrains the unlearning gradient by projecting it onto the half-space where retention is preserved, using the gradient of the loss computed on mini-batch retain samples. This framework can be applied orthogonally to common objective-based machine unlearning methods, and the authors demonstrated its removal efficacy and retention reliability with both empirical evaluation (on TOFU, WMDP and MUSE benchmarks) and theoretical analysis. The authors also explored the retain-data-free setting by incorporating task vectors, which shows performance gain on the TOFU benchmark.

**Claims And Evidence:**

In general, I find the claims supported by clear and convincing evidence.

* The paper claims that "GRU offers enhanced reliability in retention", which matches with intuition, as GRU uses the retention gradient to adjust the original unlearning gradient, ensuring no negative projection conflicting with the retention direction. This claim is supported by empirical evidence: model utility consistently improves when GRU is applied with current unlearning methods, especially for GA-based methods where the retain set was not originally used. The authors also present theoretical evidence with Theorem 3.2, showing that the retention loss with GRU does not surpass that from the original unlearning method. Contingent on its proof being sound (see the `Theoretical Claims` section for concerns), I think this claim is well-supported.

* Regarding the aspect of forget quality, the paper claims that GRU achieves “powerful unlearning capabilities” (L91). Although Theorem 3.1 and the experiments provide some evidence for this, I find it somewhat over-claiming. The improvement in removal effectiveness on WMDP appears not as evident or consistent as on TOFU (as seen in Fig. 4a and the WMDP-cyber results in Table 3). Given that the room for improvement on MUSE with the KnowMem metric is limited (Fig. 5a), it would be more convincing if the authors included other metrics introduced in MUSE (e.g. PrivLeak). Additionally, it would be helpful if more baselines can be included, such as RMU [1] and FLAT [2].


[1]. The WMDP Benchmark: Measuring and Reducing Malicious Use With Unlearning
[2]. LLM Unlearning via Loss Adjustment with Only Forget Data

**Essential References Not Discussed:**

This paper has significant overlap in terms of methodology with [4], which was published in MM '24. [4] uses the same idea of gradient projection when there is a conflict (termed "Gradient Direction Rectification"; see Sec 4, Eq 4, and Algorithm 1 of the paper). Given that this gradient projection idea is indeed intuitively simple, I do acknowledge that it is likely that the authors have developed this idea independently but missed this work unintentionally. Nevertheless, a reference and a discussion on the similarities and differences should be added.

Some less significant problems include:

* Some related baselines could be added to strengthen the paper, as mentioned in the `Claims And Evidence` section.

* WGA [5] was cited and used as a baseline method in the experiments. However, it was not discussed in Section 2.2. Notably, this paper also examines the gradient direction of the unlearning objective. A discussion clarifying the differences between the two works should also be included.

[4]. GDR-GMA: Machine Unlearning via Direction-Rectified and Magnitude-Adjusted Gradients

[5]. Rethinking LLM Unlearning Objectives: A Gradient Perspective and Go Beyond

**Experimental Designs Or Analyses:**

The provided experimental designs and analyses are valid, and the authors adhere to the default experimental designs and settings from the chosen benchmarks.

**Methods And Evaluation Criteria:**

The problem is to mitigate the tradeoff between removal and retaining in LLM unlearning, and specifically, the side-effect or excessive unlearning that is common among current gradient-based methods. The proposed methods (both GRU and TRU) make sense and is well-motivated by the intuition of projecting out the component opposite from the retain gradient direction, thereby forcing a non-obtuse angle that regulates the gradient updates, leading to a smaller retain loss and potentially better retention or minimal side-effect on model utility.

The choice of the three commonly used benchmarks is reasonable, and the authors follow the default evaluation settings for each benchmark, even though this requires using different backbones and tuning specific hyperparameters for each dataset. However, additional metrics or baselines could be included, as noted in the previous section.

**Other Comments Or Suggestions:**

Minor typos:

Theorem 3.2 Remark., "heurstics" -> "heuristics"

Section 4, paragraph 2, "eeach" -> "each"

**Other Strengths And Weaknesses:**

Strengths:

* The authors extend this idea of rectification to the retain-data-free setting, which is arguably more valuable.

Weaknesses:

* While the method is simple, intuitive, and well presented with geometric illustration, similar gradient projection ideas for unlearning has been proposed in previous works (see `Essential References Not Discussed`), which impacts its novelty and originality.
* The effectiveness of this method could be sensitive to the size, quality, and distribution of the retain set.
* There lacks analysis on the efficiency and added complexity of this method, such as wall-clock time for a gradient update step.

**Questions For Authors:**

1. How are mini-batch retain samples selected for baselines that do not have a retain set, e.g. GA and WGA?

2. For the evaluation setting of TOFU, are you using exactly the TS-test (log) p-values as FQ? And for MU, are you using only the Truth Ratio sub-metric, or the aggregation of Truth Ratio, Prob, and ROUGE-L? It would be helpful if you could clarify these details in the paper.

3. By default, MUSE used LLaMA-2-7B on the news subset and ICLM-7B for the HP books subset, and you are only using the ICLM-7B model. Are you testing on the entire MUSE, or just the books subset?

4. Could GRU be seen as an explicitly-constrained variant of GD that relies on retention gradient estimates from up-sampled retain data?

5. Would GRU/TRU be robust to relearning attacks?

**Relation To Broader Scientific Literature:**

The proposed idea addresses the problem of excessive unlearning, and therefore mitigates the fundamental tradeoff in machine unlearning between adequate removal and model utility. Some results and findings, such as GA-based methods can be greatly improved by augmenting with GRU, and NPO-based methods generally work better, align with broader scientific literature. The exploration of TRU without requiring retain data displays potential for this framework to benefit a wider range of unlearning methods.

**Theoretical Claims:**

I checked the correctness of the closed-form solution derivation in A.1 and the proof of Theorem 3.1 in A.2.

For the proof of Theorem 3.2 in A.3, the authors start by introducing the definition of q-Curvature, with which I am not familiar. Despite a careful literature search (e.g. from differential geometry [3]), I was unable to find a reference of this definition within the machine (un)learning community, or determine how this concept is derived from existing mathematical literature. Therefore, I am not able to assess the soundness of this proof.

It would be much appreciated if the authors could provide more background and specify how it is adapted from previous works.

[3]. (2010). Q-curvature. In: Conformal Differential Geometry. Oberwolfach Seminars, vol 40. Birkhäuser Basel. https://doi.org/10.1007/978-3-7643-9909-2_1

---

> ### Author Rebuttal · Authors · 2025-03-31
>
> Due to strict space limits, we try our best to address the most critical questions as briefly as possible. We sincerely welcome any further concerns and will try our best to respond to them.
> > Q1. The improvement in removal on WMDP appears not as evident or consistent as on TOFU.
>
> **A1**. WMDP uses a "Choose 1 from 4 QA accuracy," where **random guessing (totally unlearned) would result in about 0.25**. As observed, QA accuracies are already near 0.25, leaving little room for further declines. Our efficacy on WMDP is actually reflected by retention, with 2-7 percentage points increase over that without GRU. These results confirm that **GRU maintains strong unlearning while improve retention**, aligning our pirmary goal.
> > Q2. It would be helpful if more baselines can be included, such as RMU and FLAT.
>
> **A2**. Many thanks for your suggestion. We present results using RMU and FLAT on the WMDP, along with their versions with GRU. As observed, GRU indeed enhances performance for both retention and unlearning.
> Method|WMDP Bio↓|WMDP Cyber↓|WMDP MMLU↑|
> |:-:|:-:|:-:|:-:|
> |RMU|0.26|0.31|0.41|
> |w/ GRU|0.26|0.28|0.44|
> |FLAT|0.25|0.25|0.27|
> |w/ GRU|0.24|0.25|0.28|
>
> We adjust $\alpha$ from the default 1200 to 100 in the RMU open-sourced code, after finding that the original settings caused the retain term to overly dominate, hindering model updates. Similar sensitivity issues, such as precision setups, have been reported by others in the WMDP GitHub repository.
>
> We also expect the potential to further improve RMU and FLAT. For RMU, which perturbs localized representations, we aim to develop mask-based constraints for better knowledge localization. For FLAT, it uses a preference-based learning framework with distinct gradient behaviors from methods like GA and NPO. Thus, we need to devise new retain risk for more appropriate constraints, which we will explore in the future.
>
> > Q3. This paper has significant overlap in terms of methodology of GDR.  WGA also examines the gradient direction of the unlearning objective.
>
> **A3**. **Regarding GDR**, its requirement to store gradients across epochs is infeasible for LLMs due to memory costs and fewer epochs (e.g., 1 for MUSE). GRU avoids extensive caching and epoch-dependent operations, which is more practical for LLMs. Also, for LLMs, the retain data can lead to overfitting when directly guiding updates as in GDR. GRU, by not directly merging retain gradients into its updates, avoids this issue. Also, our enhanced version, TRU, explicitly address this concern, which is superior over GDR.
>
> **Regarding WGA's associated paper**, it employs gradient computations to find reliable unlearning objectives. While conceptually possible, its practical challenges can moviate our work into optimization-focused research. Thus, **our work extends beyond WGA's, further emphasizing the importance in exploring gradients, while focusing on a promising yet orthogonal direction**.
>
> > Q4. It would be much appreciated if the authors could provide details to help validate the results.
>
> **A4**. The configurations are detailed in Section 5 and Appendix C.1. The batch size for retain samples, set at 32, mirrors the unlearn batch and follows the random sampling as to GD. We further provide our source codes via the [Anonymous GitHub](https://anonymous.4open.science/r/GRU-664A/).
> > Q5. For TOFU, are you using the TS-test (log) p-values as FQ? And for MU, are you using the aggregation of Truth Ratio, Prob, and ROUGE-L?
>
> **A5**. To enhance readability and ease figure plotting, we uniformly report the $\log p$ values, avoiding dealing with extremely small $p$ like $1e^{-19}$. For MU, we adhere to TOFU, reporting the aggregated metrics.
> > Q6. By default, MUSE used LLaMA-2-7B on the news subset and ICLM-7B for the HP books subset, and you are only using the ICLM-7B model. Are you testing on the entire MUSE, or just the books subset?
>
> **A6**. Following your suggestion, we further report results on MUSE News with LLaMA-2-7B, taking GA and NPO due to space limit. UWC is adopted for calibration towards a fair comparison. As observed, there are notable improvements in PrivLeak, espeically for GA.
>
> Method|VerbMem ↓|KnowMem-U ↓|KnowMem-R ↑|PrivLeak → 0|
> |:-|:-:|:-:|:-:|:-:|
> |GA|31|60|47|-100|
> |w/ GRU|11|56|47|-18|
> |NPO|43|58|47|-100|
> |w/ GRU|35|50|47|-96|
> > Q7. Could GRU be seen as an explicitly-constrained variant of GD that relies on retention gradient estimates from up-sampled retain data?
>
> **A7**. Yes, your understanding is correct. If GA as the objective and combining it with GRU corresponds to the explicit-constrained variant of GD. The benefits of our approach are clear, which forcibly ensures retention. In contrast, using original GD can result in unlearning gradients dominating updates, adversely affecting retention.

---

> > ### Comment · Reviewer_z9sW · 2025-04-03
> >
> > Thank you for your reply and additional experiments, which addressed some of my concerns. However, I think a critical issue remains:
> >
> > * Regarding originality, novelty, and comparison between previous works: GDR and this work share the same core idea of gradient rectification, which projects the task gradient onto the orthonormal plane of the conflicting gradient. Similar gradient projection idea, I should note, has been widely adopted in multi-task learning literature when two or more tasks are in trade-offs. Specifically, Gradient Surgery/PCGrad [1] used the same projection rule. It is also worth noting that Theorem 3.1 (convergence guarantee) and 3.2 (loss improvement guarantee) share great similarity with Theorem 1, Definition 5 and Theorem 2, 3 of [1].
> >
> >     To be clear, I do think it's a fair contribution to adapt similar ideas to LLMs and LLM unlearning (where the two tasks, forget and retain, can be in conflict), but essential citations and discussions should be included.
> >
> > [1]. Gradient Surgery for Multi-Task Learning (2020; citations > 1200)
> >
> > &nbsp;
> >
> > Some further questions:
> >
> > 1.
> > > Regarding GDR, its requirement to store gradients across epochs is infeasible for LLMs due to memory costs and fewer epochs (e.g., 1 for MUSE). GRU avoids extensive caching and epoch-dependent operations, which is more practical for LLMs.
> >
> >     Can you elaborate on how you did that? From the code (`dataloader.py`), it appears that you are still caching the unlearning and retain gradients in `compute_loss` for every training step, flatten and store the structure map of each gradient with `flatten_and_store_grads`. To me, this does not seem less complex than the [GDR implementation](https://github.com/RUIYUN-ML/GDR-GMA/blob/main/memory_bank.py).
> >
> > 2. How often is the dot similarity negative (thus requiring gradient projection/adjustment)? Does the frequency of it being negative and its magnitude vary across epochs, or perhaps dependent on tasks/datasets (e.g. degree of overlap between retain and forget set)?
> >
> > I will consider update my score if the above concerns are well-addressed.
> >
> > ---
> >
> > **Edit**:
> >
> > I appreciate the frank and engaging response, and the ablation regarding EMA. Contingent on acknowledging the scope as "adapting existing gradient rectification techniques to LLM unlearning" and including a discussion on related work, I'm OK with this paper. I have raised my score to 3.

---

> > > ### Author Response · Authors · 2025-04-03
> > >
> > > Thank you for your thorough and careful review! We are so happy to have engaged with a rigorous and insightful reviewer like you, whose pointed yet meaningful questions not only play a crucial role in improving the quality of this paper but also contribute a lot to the rigor and professionalism of the ICML community. Kindly please see our responses for your questions below.
> > >
> > > > Q1. Regarding originality, novelty, and comparison between previous works.
> > >
> > > **A1**. We totally agree that gradient rectification is a proven strategy for resolving conflict goals (e.g., in continuous learning [1] and multi-task learning, as you suggested) and acknowledge our alignment with this principle for balancing unlearning and retention. However, we remain confident in our contribution, given that **this work pioneers the adaptation of gradient rectification to LLM unlearning**.
> > >
> > > To support this claim, it is essential to highlight that, between classical machine unlearning (explored by GDR) and LLM unlearning, **the core distinction lies in retention data**: Classical machine unlearning assumes closed-world discriminative models with well-defined (in-distribution) retention distribution. For LLMs with general purpose, such a precise definition is unattainable because 1) their multi-phase training procedures (pre-training, SFT, and RL) without associated data access. Even having these data, 2) their extreme scales also makes it impracical to revisit all data during unlearning. Therefore, the retention data adopted for LLM unlearning are actually surrogate and biased.
> > >
> > > This factor motivates our paper to be structured as follows: In Section 3, we first show the possibility of incorporating gradient rectification into LLM unlearning, leading to GRU. Then, we delve into the limitations of current LLM unlearning setups, proposing TRU to better tackle the issue of ill-defined (or biased) retention data. TRU is a preliminary step toward more reliable LLM unlearning, with future work planned to expand its scope.
> > >
> > > We will carefully highlight these crucial points in our revision. We will **add a section on related work**, where we will review the literature on existing gradient rectification techniques and explore the differences between classical machine unlearning and LLM unlearning. Moreover, for the theoretical derivation, we recognize the similarities with PCGrad. We will incorporate a detailed discussion and extra remarks to further highlight their contributions. We would like to express our sincere thanks again for your comments, which are critical to improving the quality and rigor of this manuscript.
> > >
> > > [1] Gradient Episodic Memory for Continual Learning. 2017.
> > >
> > > > Q2. From the code, it appears that you are still caching the unlearning and retain gradients in compute_loss for every training step, flatten and store the structure map of each gradient with flatten_and_store_grads.
> > >
> > > **A2**. ``flatten_and_store_grads`` is used to reduce GPU memory usage during gradient rectification (preventing out-of-memory issues). Its values will be cleaned at the end of each step and does not function as a gradient bank. Below, we would like to further highlight the key differences in the implementations between GRU and GDR.
> > >
> > > In our GRU, we cache an exponential moving average (EMA) to achieve a more accurate estimation of the **average retain gradients**, which are dynamically updated. It is implemented as follows:
> > > ```
> > > self.flattened_retain = self.moving_avg * self.flattened_retain_accumulation + (1 - self.moving_avg) * self.flattened_retain_old
> > > ```
> > >
> > > On the other hand, GDR requires storing each **individual batch of gradients**, as indicated by the following line of code:
> > > ```
> > > bank = MemoryBank(size=math.ceil(t_dataset_sizes/args.batch_size))
> > > ```
> > >
> > > Overall, the memory cost for our GRU is equivalent to the memory required for storing parameters that necessitate gradients. In contrast, GDR further multiplies this cost by the number of unlearning steps within each epoch. Therefore, a clear advantage of our approach is that **we notably reduce the additional memory costs compared to GDR**, making our method more practical for LLMs. Moreover, our GRU maintains reliable performance even without caching. We conduct experiments below to validate this claim, where GRU is implemented without EMA.
> > >
> > > Method |FQ 5%↑|MU 5%↑|FQ 10%↑|MU 10%↑|
> > > |:-:|:-:|:-:|:-:|:-:|
> > > |GA|-16.93|0.00|-14.37|0.00|
> > > |w/ GRU|-3.52|0.51|-7.34|0.22|
> > > |NPO|-10.91|0.49|-8.70|0.29|
> > > |w/ GRU|-9.96|0.54|-4.12|0.35|
> > > |GD|-13.48|0.55|-13.92|0.39|
> > > |w/ GRU|-12.42|0.56|-10.91|0.53|
> > >
> > > > Q3. How often is the dot similarity negative (thus requiring gradient projection/adjustment)?
> > >
> > > **A3**. Figure 6 in Appendix D visualizes the dot similarity for TOFU, which remains negative during unlearning. Extra visualizations have been conducted for MUSE and WMDP, as detailed in the provided [link](https://anonymous.4open.science/r/GRU-664A/cos.png), leading to the same conclusion.

---

### Official Review · Reviewer_XYtU · 2025-03-15

**Overall Recommendation:** 4

**Summary:**

The authors introduce GRU as a flexible framework designed to be integrated with existing unlearning methods to balance the trade-off between knowledge removal and retention in LLM unlearning. GRU constrains unlearning gradients to minimize their negative impact on retention. Theoretical analysis and empirical evaluations across multiple benchmarks and diverse unlearning baselines confirm its effectiveness.

**Claims And Evidence:**

The authors present a detailed theoretical analysis explaining how GRU effectively mitigates the negative impact on retention. Additionally, extensive experimental results further validate its effectiveness.

**Essential References Not Discussed:**

The authors have cited most related works.

**Ethical Review Flag:**

Flag this paper for an ethics review.

**Experimental Designs Or Analyses:**

The authors present detailed experimental settings, evaluations, and baselines. The design and analysis are well-structured and valid.

**Methods And Evaluation Criteria:**

The authors incorporate the proposed GRU into several well-established baselines, including GA, WGA, NPO, and GD, and conduct extensive experiments on widely used benchmarks such as TOFU, WMDP, and MUSE.

**Other Comments Or Suggestions:**

1. The preliminaries section takes up excessive space. The authors should consider making this section more concise while ensuring it remains comprehensive.

**Other Strengths And Weaknesses:**

Strengths

1. The authors use two figures, Figure 1 and Figure 2, to effectively illustrate the motivation and functionality of GRU with clarity.
2. In addition to GRU, the authors introduce Task Vector Rectified Unlearning (TRU) to remove the dependency on a retention set.

Weaknesses

1. Some concepts lack clarity. For example, it is unclear how the constraints on gradients from different samples and mini-batches (i.e., the random mini-batch from $D_r$ and the mini-batch from $D_u$)  are enforced in Eq. (7). The authors should provide further explanation or justification.
2.  There are too many critical hyper-parameters, such as $\gamma$ and $\tau$. Performing grid search to find the optimal values for these hyper-parameters is computationally expensive and time-consuming.
3. Some claims need further justification, for example, "for each individual data point $s_u \in D_u$ targeted for unlearning, the remaining data points within $D_u$, i.e.,$D_u$ \ ${s_u}$, can offer information for retention if used properly.".
4. The experimental results presented in the figures, i.e., Figure 3 and Figure 4, are not clear.

**Questions For Authors:**

1. You mentioned that "retain data can often be biased." Could you clarify what bias refers to in this context? Does it relate to dataset distribution shifts, representation imbalances, or another specific aspect? A more precise explanation would improve clarity.
2. See weaknesses in the above section.

**Relation To Broader Scientific Literature:**

This paper introduces a novel regularization method that contributes to the research on LLM unlearning.

**Theoretical Claims:**

The authors present two theorems, Theorem 3.1 and Theorem 3.2, along with detailed proofs and analyses to support their validity.

---

> ### Author Rebuttal · Authors · 2025-03-31
>
> Many thanks for your great support and constructive comments! Please see our responses below.
>
> > Q1. It is unclear how the constraints on gradients from different samples and mini-batches are enforced in Eq. (7).
>
> **A1**. As shown in Algorithm 1, **mini-batches $B_{\rm r}$, $B_{\rm u}$ replace $D_{\rm r}$, $D_{\rm u}$**. In Section 3.2, we further discuss the limitations of mini-batch gradients and recommend using the exponenital moving average for stable estimation. We will clarify these points in our revised manuscript. Sincere thanks for your suggestion!
>
> > Q2. Performing grid search to find the optimal values for these hyper-parameters is computationally expensive and time-consuming.
>
> **A2**. Many thanks for raising this concern. We would like to clarify the practicality of our approach from the following two perspectives.
>
> 1. **The GRU performs well even without these hyper-parameters.** Our basic framework following Eq. (7) operates without any hyper-parameters. Incorrperating $\gamma$ (for EMA) and $\tau$ (for clipping) are practical enhancements that improve empirical results. Below is the table showing results for 3 methods under the LLaMA setup, without using $\gamma$ and $\tau$. The results highlight notable improvements of our method over the baselines.
>
> Method |FQ 5%↑|MU 5%↑|FQ 10%↑|MU 10%↑|
> |:-:|:-:|:-:|:-:|:-:|
> |GA|-16.93|0.00|-14.37|0.00|
> |w/ GRU|-3.52|0.51|-7.34|0.22|
> |NPO|-10.91|0.49|-8.70|0.29|
> |w/ GRU|-9.96|0.54|-4.12|0.35|
> |GD|-13.48|0.55|-13.92|0.39|
> |w/ GRU|-12.42|0.56|-10.91|0.53|
>
> 2. **The GRU is not very sensitive to these hyper-parameters.** By incorporating $\gamma$ and $\tau$, we explore the potential to further enhance unlearning. In Appendix D, we conduct a hyper-parameter sensitivity analysis, showing stable improvements across a wide range of candidate hyper-parameters.
>
> > Q3. Some claims need further justification, for example, "..., the remaining data points within  $D_{\rm u}$ can offer information for retention.".
>
> **A3**. Heuristically, retain gradients primarily **act as a denoising mechanism, rather than directly contributing update directions**. By using retain gradients to redirect original unlearn gradients, we focus on the direction that removes targeted knowledge, while with limited side impacts on unrelated knowledge.
>
> For TRU, let's first consider a simple scenario to remove a single $s_{\rm u}$. Here, it is straightforward to take the complementary, i.e., $D_{\rm u} \backslash{s_{\rm u}}$, for retention. We then create a rectified task vector targeted at eliminating knowledge associated with $s_{\rm u}$ while minimizing impact on unrelated knowledge. This process can be applied to each $s_{\rm u}$ within $D_{\rm u}$, with the collective average forming the rectified task vector for the entire $D_{\rm u}$.
>
> Notably, **these task vectors remain mutally compatible**, as the retain gradients are exclusively employed for denoising rather than direct used into the task vectors. We will add the related discussion in our revision.
>
>
> > Q4. The results presented in the figures are not clear.
>
> **A4**. Apologies for any confusion caused by figure annotations. As shown in captions, each pair of scores represents metric values (either FQ or MU) before and after applying GRU. Taking Figure 3(a) as an example, the pair (-16.93, -3.52) represents FQ for GA, where -16.93 is the score without GRU and -3.52 is the score with GRU. The visual representation using an upward growing grid between these scores emphasizes the achieved improvement. We will provide more  explanations in our revision to ensure clarity.
>
>
> > Q5. The preliminaries section takes up excessive space. The authors should consider making this section more concise while ensuring it remains comprehensive.
>
> **A5**. We have a relatively extensive preliminaries section to ensure self-containment and maintain clarity. However, we fully concur with your opinion that a more concise verision would enhance the paper flow and allow larger space to elaobrate on our core contributions regarding GRU and TRU. We sincerely appreciate your feedback and will carefully refine our paper structure accordingly.
>
> > Q6. You mentioned that "retain data can often be biased." Could you clarify what bias refers to in this context?
>
> **A6**. Many thanks for your question. Taking TOFU as an example, the current setup involves selectively unlearning specific author profiles while retaining others. However, we recognize that the broader objective of retention should ensure model capacity across diverse domains such as humanities and sciences. Therefore, the current retain data in TOFU may exhibit bias, **stemming from the distribution shift between the adopted retain data and the broader real data**. We will refine the related discussion in our revision.

---

### Decision · Program_Chairs · 2025-05-01

**Decision:**

Accept (poster)

**Comment:**

This paper addresses an important and timely challenge in LLM unlearning by proposing Gradient Rectified Unlearning (GRU), a simple yet effective technique that leverages gradient projection to reduce negative side effects on retained knowledge. Reviewers generally agree that the paper is well-written and addresses a critical trade-off in unlearning. The main concern across reviewers is limited originality, as the core idea of gradient projection has precedents in multi-task learning and recent unlearning works. Some reviewers found the initial omission of key related work and baselines to be a weakness. The authors provided thorough and thoughtful rebuttals, acknowledged prior works, and clarified implementation and evaluation details, together with additional experiments in addressing major concerns. All reviewers updated or maintained a positive score, indicating a consistent lean toward acceptance.